# ATOMAS: HIERARCHICAL ADAPTIVE ALIGNMENT ON MOLECULE-TEXT FOR UNIFIED MOLECULE UNDERSTANDING AND GENERATION

**Yikun Zhang**[2*], **Geyan Ye**[1†,*] **Chaohao Yuan**[4], **Bo Han**[5], **Long-Kai Huang**[1], **Jianhua Yao**[1], **Wei Liu**[1], **Yu Rong**[3] [†]
[1]Tencent AI Lab [2]Peking University [3]DAMO Academy, Alibaba Group [4]Tsinghua University
[5]Hong Kong Baptist University
{yikun.zh,yu.rong}@hotmail.com, blazerye@tencent.com

## ABSTRACT

Molecule-and-text cross-modal representation learning has emerged as a promising direction for enhancing the quality of molecular representation, thereby improving performance in various scientific fields. However, most approaches employ a global alignment approach to learn the knowledge from different modalities that may fail to capture fine-grained information, such as molecule-and-text fragments and stereoisomeric nuances, which is crucial for downstream tasks. Furthermore, it is incapable of modeling such information using a similar global alignment strategy due to the lack of annotations about the fine-grained fragments in the existing dataset. In this paper, we propose Atomas, a hierarchical molecular representation learning framework that jointly learns representations from SMILES strings and text. We design a Hierarchical Adaptive Alignment model to automatically learn the fine-grained fragment correspondence between two modalities and align these representations at three semantic levels. Atomas's end-to-end training framework supports understanding and generating molecules, enabling a wider range of downstream tasks. Atomas achieves superior performance across 12 tasks on 11 datasets, outperforming 11 baseline models thus highlighting the effectiveness and versatility of our method. Scaling experiments further demonstrate Atomas's robustness and scalability. Moreover, visualization and qualitative analysis, validated by human experts, confirm the chemical relevance of our approach. Codes are released on https://github.com/yikunpku/Atomas.

## 1 INTRODUCTION

Molecular representation learning is crucial in fields like drug discovery (Drews, 2000; Liu et al., 2023b), virtual screening (Walters et al., 1998; Goel et al., 2023), and molecular design (Ye et al., 2023; Thomas et al., 2023). Recent advances in molecule-and-text cross-modal models (Liu et al., 2023c; Luo et al., 2023; Liu et al., 2023e) have enhanced the generalization of molecular representations by integrating internal structures (SMILES strings, structural data) and external domain knowledge (textual descriptions, knowledge graphs).

However, current approaches encounter three primary challenges. **(1) Fine-Grained Correspondence:** Existing molecule-and-text alignment methods (Christofidellis et al., 2023; Edwards et al., 2022; Liu et al., 2023c; Luo et al., 2023; Liu et al., 2023e) struggle to effectively capture fine-grained correspondence related to local parts within different modalities, which is essential for downstream molecular tasks (Xia et al., 2023). For example, molecule captions generated by global alignment often fail to distinguish between 'D-glutamate' and 'L-glutamate' enantiomers, indicating a lack of sensitivity to subtle details. This oversight can lead to inaccuracies in chemical analysis and interpretation. Currently, there is a lack of fine-grained datasets with explicit annotations of local correspondences between molecules and text. Acquiring such datasets is difficult due to the com-

---

[*]Equal contributions.
[†]Yu Rong and Geyan Ye are the corresponding authors.

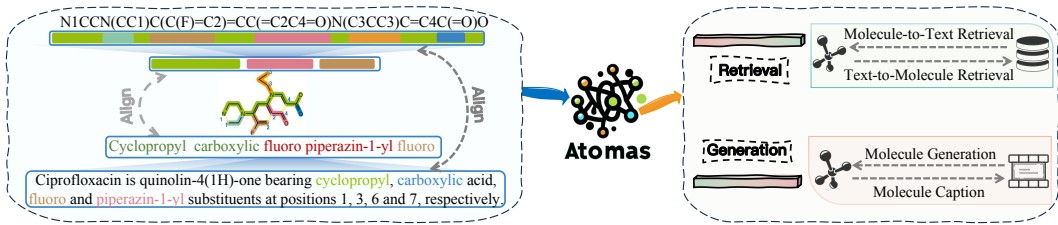

Figure 1: Atomas is a hierarchical, end-to-end model designed to discover and automatically align local substructures of input while performing conditional generation. The learned cross-modal representations can be adapted to both understanding tasks (retrieval tasks) and generation tasks.

plexity and specialization of molecular strings and text descriptions, which require extensive human expert annotation. This limitation hinders the ability of globally aligned methods to effectively learn fine-grained information and address fine-grained alignment challenges. **(2) Molecular Modality Focus:** Current fine-grained alignment methods (Feng et al., 2023; Ji et al., 2022) focus on molecular modality and fine-grained aligning substructures. The cross-modality alignment molecular fragments and textual descriptions is largely overlooked. Existing segmentation tools for text and SMILES struggle with complexity and specialization, making it challenging to construct hierarchical text-molecular pairs. **(3) Generative Task Optimization:** Most approaches (Feng et al., 2023; Yu et al., 2024; Ji et al., 2022; Liu et al., 2022) are designed for prediction tasks and do not optimize aligned representations for generative tasks.

To this end, we propose Atomas, a hierarchical cross-modal molecular representation learning framework that jointly learns representations from SMILES and text. Figure 1 provides a conceptual illustration of Atomas. In Atomas, we exploit the unique characteristic of SMILES as a specialized form of text and employ a unified encoder for both SMILES and text modalities. This results in more isomorphic representations for both modalities, thereby facilitating subsequent alignment tasks. Meanwhile, considering the textual description of molecules naturally has a hierarchical structure and the need for local molecular alignment, we design a Hierarchical Adaptive Alignment (HAA) model which comprises two components: Adaptive Polymerization Module (APM) and Weighted Alignment Module (WAM). APM assigns the quantities of the low-level tokens into high-level tokens (fragments) for the single modality, while WAM leverages token representations from both SMILES and text modalities to automatically learn the matching of token and aligns the representation of two modalities in a set-wise manner. By iteratively invoking APM and WAM, we devised a three-level alignment scheme (atom level, fragment level, and molecule level). This hierarchical alignment structure enables improved learning of local alignment across different abstraction levels within the two modalities. Additionally, by incorporating a conditional decoder within the alignment process, Atomas can optimize the representation of molecule and text specifically for generation tasks. Our contributions are summarized as follows:

- To the best of our knowledge, Atomas is the pioneering molecule-and-text representation learning framework that tackles the challenge of aligning local information without the need for explicit labeling between text fragments and molecular substructures.

- We introduce the concept of Hierarchical Adaptive Alignment, enabling automatic learning the fine-grained correspondence between molecule and text at three semantic levels from coarse to fine. Atomas achieves state-of-the-art performance on a wide range of molecule-text tasks, including molecule and text retrieval, text-based de novo molecule generation, and molecule captioning.

- Atomas brings new insights into molecule generation tasks: (1) Aligning before generation improves the efficacy of molecule conditional generation tasks. (2) Fine-grained hierarchical alignment enhances the quality of controllable molecule generation. (3) Joint optimization within a unified training framework surpasses the efficacy of a two-stage approach for molecular generation tasks. (4) Employing a unified encoder may advantageous in scenarios characterized by data scarcity.

## 2 RELATED WORKS

**Molecule-and-Text Cross-Modal Models:** (Edwards et al., 2022; Christofidellis et al., 2023; Liu et al., 2023f;c; Luo et al., 2023; Chen et al., 2024) investigate the interaction between molecular

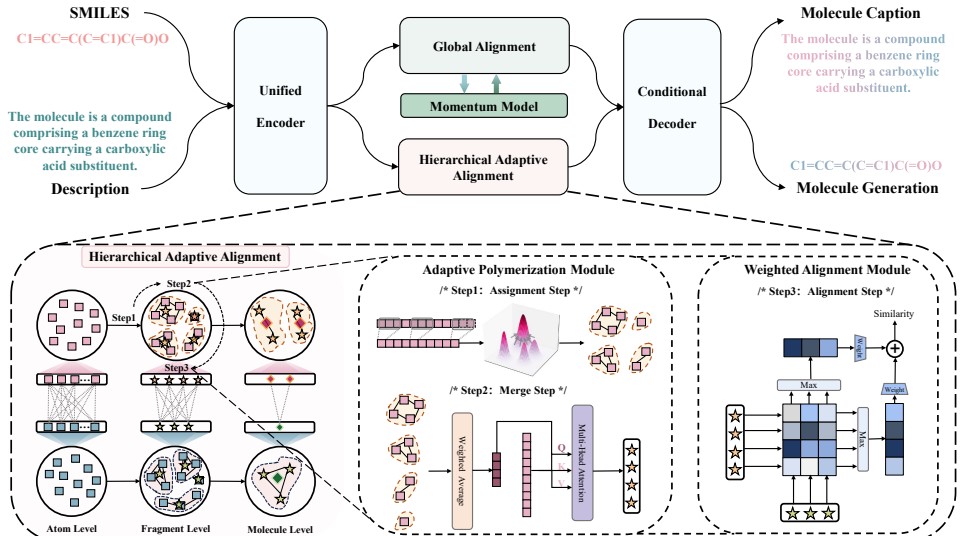

Figure 2: **Illustration of the proposed Atomas.** Atomas is composed of four components. (1) Unified Encoder encodes both the input molecule and its corresponding textual description. (2) Global Alignment module projects and aligns the global features of the molecule and text. A momentum model is used to ensure alignment consistency. (3) Hierarchical Adaptive Alignment aligns the molecule and text at three levels, including the Adaptive Polymerization module which clusters the original token features into distinct representation sets, and the Weighted Alignment module which aligns two modalities in a set-wise manner. (4) Conditional Decoder takes the molecule and text embedding as input and generates the target modality.

structures and textual descriptions to complement and enhance the overall information content. However, these methods only consider global representations of SMILES and text, overlooking finer-grained modal interactions. Atomas introduces hierarchical fine-grained alignment between SMILES and text, which is crucial for controlled molecule generation and molecular captioning, enabling better performance. While (Edwards et al., 2022; Christofidellis et al., 2023) treat conditional generation tasks as translation tasks without establishing alignment, Atomas demonstrates the efficacy of performing preliminary alignment. Unlike (Liu et al., 2023f;c; Luo et al., 2023; Chen et al., 2024)'s two-stage training strategies, Atomas employs a joint optimization approach, which is essential for effectively learning and generating molecular representations. More discussions are provided in Appendix A.1.

**Molecular Modalities Representation Learning:** (Feng et al., 2023; Yu et al., 2024) focus on fine-grained aligning molecular modalities and tailoring for prediction tasks. Different from these paradigms, Atomas addresses cross-modal learning between molecule and text modalities. Challenges arise from the lack of expert fine-grained textual annotations for molecules and difficulty in constructing positive and negative pairs, as a text fragment may suit multiple molecule substructures. These challenges make Atomas' achievements in this field particularly noteworthy. See Appendix A.2 for more discussions.

## 3 METHODOLOGY

In this section, we present the details of each component of Atomas. Figure 2 illustrates the overall framework of Atomas. The algorithm details are shown in Appendix D.

### 3.1 MOLECULE-TEXT UNIFIED ENCODING

For a SMILES-text pair $M = (S, T)$, SMILES $S$ and text description $T$ are fed into the unified encoder $f_\theta$ since both are essentially languages. We discuss the advantages of utilizing a unified encoder in Section 4.6. The input $T$ with $N_{ta}$ tokens is embedded into word sequence $\boldsymbol{T}_a = \left\{\boldsymbol{t}_a^i\right\}_{i=1}^{N_{ta}}$,

where $\boldsymbol{t}_a^i \in \mathbb{R}^{D_t}$ denotes the feature vector of the $i$-th word. The input $S$ with $N_{sa}$ tokens is embedded into atom sequence $\boldsymbol{S}_a = \left\{\boldsymbol{s}_a^j\right\}_{j=1}^{N_{sa}}$, where $\boldsymbol{s}_a^j \in \mathbb{R}^{D_s}$. Since the T5 model does not have the [CLS] token, we first aggregate the $\boldsymbol{T}_a$ and $\boldsymbol{S}_a$ by a projection module $proj\left(\cdot\right)$ to obtain the global feature $\boldsymbol{t}_g$ and $\boldsymbol{s}_g$ :

$$\boldsymbol{t}_g = proj(\boldsymbol{T}_a) = \boldsymbol{W}_t^\top \boldsymbol{T}_a + \boldsymbol{b}_t, \quad \boldsymbol{s}_g = proj(\boldsymbol{S}_a) = \boldsymbol{W}_s^\top \boldsymbol{S}_a + \boldsymbol{b}_s, \tag{1}$$

where $\boldsymbol{W}_t \in \mathbb{R}^{N_t \times 1}$ and $\boldsymbol{W}_s \in \mathbb{R}^{N_s \times 1}$ are the learned matrix, $\boldsymbol{b}_t$ and $\boldsymbol{b}_s$ are the bias terms.

We then align the global representation pair $(\boldsymbol{t}_g, \boldsymbol{s}_g)$ by performing cross-modal contrastive learning. To ensure sufficient negative pairs and consistent feature representation from both modalities, we follow (He et al., 2020) to introduce a momentum unified encoder $f_\theta^m$ and two queues for text $\boldsymbol{t}$ and SMILES $\boldsymbol{s}$ denote as $\boldsymbol{Q}_t$ and $\boldsymbol{Q}_s$, respectively. $f_\theta^m$ is updated by $f_\theta$ in the following way,

$$f_\theta^m \leftarrow \alpha f_\theta^m + (1 - \alpha) f_\theta, \tag{2}$$

where $\alpha \in [0, 1)$ is a momentum coefficient parameter and only the parameters $f_\theta$ are updated by back-propagation.

$\boldsymbol{Q}_t$ and $\boldsymbol{Q}_s$ store the global feature $\boldsymbol{t}_g^{'}$ and $\boldsymbol{s}_g^{'}$ generated by $f_\theta^m$, thereby creating two large and consistent dictionaries that cover a rich set of negative samples. By doing this, we calculate the global similarity score of text-to-SMILES within the specified queue range instead of in a mini-batch:

$$\boldsymbol{S}_g(\boldsymbol{t}, \boldsymbol{s}^{'}) = \frac{\exp(sim(\boldsymbol{t}_g, \boldsymbol{s}_g^{'})/\tau)}{\sum_{q=1}^Q \exp(sim(\boldsymbol{t}_g, \boldsymbol{Q}_s^q)/\tau)}, \tag{3}$$

where $\tau$ is a learnable temperature parameter. and $sim(\cdot, \cdot)$ is the similarity metric, here we calculate it using the cosine similarity function. Similarly, we can obtain the global SMILES-to-text similarity score $\boldsymbol{S}_g(\boldsymbol{s}, \boldsymbol{t}^{'})$. Inspired by (Li et al., 2021; 2022), soft labels are created from the momentum encoder $f_\theta^m$ as training targets to account for the potential positives in the negative pairs. Then the global alignment loss $\mathcal{L}_{ga}$ can be formulated as:

$$\mathcal{L}_{ga} = -\frac{1}{2}\left\{[(1-\beta)\boldsymbol{y}^{t2s} + \beta \boldsymbol{S}_g(\boldsymbol{t}^{'}, \boldsymbol{s}^{'})]\log(\boldsymbol{S}_g(\boldsymbol{t}, \boldsymbol{s}^{'})) + [(1-\beta)\boldsymbol{y}^{s2t} + \beta \boldsymbol{S}_g(\boldsymbol{s}^{'}, \boldsymbol{t}^{'})]\log(\boldsymbol{S}_g(\boldsymbol{s}, \boldsymbol{t}^{'}))\right\}, \tag{4}$$

where $\beta$ is a hyperparameter controlling label smoothness. $\boldsymbol{y}^{s2t}$ and $\boldsymbol{y}^{t2s}$ denote ground-truth similarity, with negative pairs assigned a probability of 0 and positive pairs a probability of 1.

## 3.2 HIERARCHICAL ADAPTIVE ALIGNMENT

Given an encoded SMILES-text pair $\boldsymbol{M} = (\boldsymbol{S}, \boldsymbol{T})$, it is challenging to explicitly extract the corresponding fine-grained information (*e.g.*, functional groups in SMILES and phrases in text) from $\boldsymbol{S}$ and $\boldsymbol{T}$. To address this, we propose an **adaptive polymerization module** that clusters token-wise features into disentangled representation sets. Subsequently, we introduce a **weighted alignment module** to estimate the correlation between the two modalities and identify potential active units in a set-wise manner. Figure 2 illustrates the framework of hierarchical adaptive alignment. The adaptive polymerization module includes an assignment step and a merge step. The weighted alignment module performs the alignment step. By stacking these two modules, we expand the fine-grained alignment between SMILES and text to the hierarchical interaction. The effectiveness of this hierarchical adaptive alignment is demonstrated in Table 7.

Specifically, we perform hierarchical adaptive alignment at three levels: **atom level**, where atom is aligned with word; **fragment level**, where functional group is aligned with phrase; and **molecule level**, where molecule is aligned with paragraph. Thus, it process alternates between three steps in a level-wise manner: assignment step, merge step, and alignment step.

**Assignment Step:** We utilize a learnable token aggregation module to implement adaptive polymerization. The density peak-based clustering algorithm with k-nearest neighbors (Du et al., 2016; Yao et al., 2019; Zeng et al., 2022a; Jin et al., 2023) (DPC-KNN) is utilized to assign tokens to clusters. Starting with the atom level, we firstly take atom (word) token features $\boldsymbol{S}_a = \left\{\boldsymbol{s}_a^j\right\}_{j=1}^{N_{sa}}$ into one-dimensional convolution to extract local features.

$$\boldsymbol{S}_a = \text{LayerNorm}(\text{Conv}(\boldsymbol{S}_a, \boldsymbol{W}, \boldsymbol{b}) + \boldsymbol{S}_a). \tag{5}$$

Then we compute the local density $\rho$ of each atom token feature $s_a^j$ according to it's k-nearest neighbors:

$$\rho_j = \exp\left(-\frac{1}{k} \sum_{s_a^i \in \mathrm{KNN}\left(s_a^j\right)} \frac{\left\|s_a^i - s_a^j\right\|_2^2}{\sqrt{D_s}},\right) + \epsilon, \tag{6}$$

where $s_a^i$ and $s_a^j$ are their corresponding SMILES token features. $D_s$ is the channel number of SMILES token features. $\mathrm{KNN}\left(s_a^j\right)$ denotes the k-nearest neighbors of an atom token $j$. $\epsilon$ is a random noise that is randomly sampled from the uniform distribution within the interval [0,1], ensuring that no tokens have the same density.

Then, we calculate the distance indicator $\delta$ for each token feature $s_a^j$ by determining the minimum distance between it and any other token possessing a higher local density. As for the token with the highest local density, its indicator is determined as the maximum distance between it and any other tokens:

$$\delta_j = \begin{cases} \min_{i:\rho_i > \rho_j} \left\|s_a^i - s_a^j\right\|^2, & \text{if } \exists i \text{ s.t. } \rho_i > \rho_j \\ \max_i \left\|s_a^i - s_a^j\right\|^2, & \text{otherwise.} \end{cases} \tag{7}$$

Here, $\rho$ serves as an indicator of the local density of tokens, which reflects the number of tokens located in the vicinity of $s_a^j$. $\delta$ represents the distance of a token from other high-density tokens, which measures how far it is from other tokens that are also located in highly dense regions. Together, $\rho$ and $\delta$ provide valuable information about the distribution and proximity of $s_a^i \in S_a$.

We identify tokens with relatively high values of $\rho \times \delta$ as cluster centers and then assign all other tokens to their nearest cluster center based on the Euclidean distance $d$. This clustering approach enables us to decode the input tokens into coherent semantic units, providing a more structured and meaningful representation for both word sequence $T_a$ and atom sequence $S_a$.

**Merge Step:** Tokens with similar semantic meanings may not have equal importance, so in the merge step we first assign a weight to each token feature and calculate the weighted average token features of each cluster to represent the corresponding cluster:

$$S_m^j = \frac{\sum_{k=1}^{N_{sf}^j} w_k S_a^k}{\sum_{k=1}^{N_{sf}^j} w_k}, \tag{8}$$

where $w = \mathrm{MLP}_\omega(S_a)$ is the weight of each token feature in $S_a$, $N_{sf}^j$ represents the number of features within the j-th cluster at the fragment level, $S_a^k$ is the $k$-th token feature of $S_a$ and $w_k$ is the corresponding weight score. $S_m^j$ is the $j$-th weighted average token feature.

Then we apply an attention mechanism on merged token features. $S_m$ are used as queries $Q$ and key $K$, value $V$ corresponding to the original token features $S_a$. We take the resulting output of the attention module as a higher semantic level features, i.e., functional group sequence $S_f = \left\{s_f^j\right\}_{j=1}^{N_{sf}}$, where $s_f^j \in \mathbb{R}^{D_s}$. Perform the same operation of the above two steps on word tokens to get phrase sequence $T_f = \left\{t_f^i\right\}_{i=1}^{N_{tf}}$, where $t_f^i \in \mathbb{R}^{D_t}$, $N_{tf}$ represents the number of clusters formed by word tokens. We repeat this process at the fragment level to obtain the molecule-level features.

**Alignment Step:** After the assignment and merge steps, tokens are polymerized into semantic units. we perform the weighted alignment module (Wang et al., 2022) on each level in set-wise between the SMILES and text to get the weighted average maximum alignment score. Starting with the atom level, we can obtain the text-to-SMILES similarity score:

$$S_{haa}^a(t, s) = \sum_{i=1}^{N_{ta}} w_t^i(T_a) \max_{j=1}^{N_{sa}} a_{ij}, \quad a_{ij} = \frac{(t_a^i)^\top s_a^j}{\left\|t_a^i\right\|_2 \left\|s_a^j\right\|_2}, \tag{9}$$

where $w_t = \mathrm{Softmax}(\mathrm{MLP}_{\omega_t}(T_a))$ represent the learnable weights for each textual token. The normalized alignment score $a_{ij}$ captures the similarity between the $i$-th description token feature and $j$-th SMILES token feature. Then the hierarchical adaptive alignment loss at the atom level can be calculated as:

$$\mathcal{L}_{haa}^a = -\frac{1}{2}\Big[\frac{1}{B}\sum_k^B \log\frac{\exp(\boldsymbol{S}_{haa}^a(\boldsymbol{t}_k, \boldsymbol{s}_k)/\tau)}{\sum_l^B \exp(\boldsymbol{S}_{haa}^a(\boldsymbol{t}_k, \boldsymbol{s}_l)/\tau)} + \frac{1}{B}\sum_k^B \log\frac{\exp(\boldsymbol{S}_{haa}^a(\boldsymbol{s}_k, \boldsymbol{t}_k)/\tau)}{\sum_l^B \exp(\boldsymbol{S}_{haa}^a(\boldsymbol{s}_l, \boldsymbol{t}_k)/\tau)}\Big], \quad (10)$$

where $B$ is the batch size and $\tau$ is the temperature hyperparameter. The same alignment operation is performed on the fragment level and molecule level to get $\mathcal{L}_{haa}^f$ and $\mathcal{L}_{haa}^m$.

### 3.3 CONDITIONAL GENERATION BASED ON ALIGNED REPRESENTATION

We employ a conditional generation approach to generate the target modality based on the aligned representations denoted as $\tilde{\boldsymbol{T}}_a = \big\{\tilde{\boldsymbol{t}}_a^i\big\}_{i=1}^{N_{ta}}$ and $\tilde{\boldsymbol{S}}_a = \big\{\tilde{\boldsymbol{s}}_a^j\big\}_{j=1}^{N_{sa}}$. In text-based molecule generation task, the decoder takes an aligned textual description $\tilde{\boldsymbol{T}}_a$ as input. The decoder then iteratively attends to previously generated tokens $\hat{s}_a^{<j}$ via self-attention and input condition $\tilde{\boldsymbol{T}}_a$ via cross-attention. Using these attended representations, the decoder predicts the probability of future SMILES tokens $P(\hat{s}_a^j|\hat{s}_a^{<j}, \tilde{\boldsymbol{T}}_a)$. Then the decoder can be optimized by minimizing the negative log-likelihood of label SMILES $\boldsymbol{s}$ tokens given textual description $\tilde{\boldsymbol{T}}_a$ and the same operation is applied to the molecule captioning task:

$$\mathcal{L}_{lm} = -\sum_{j=1}^{N_{sa}} \log P(\hat{s}_a^j|\hat{s}_a^{<j}, \tilde{\boldsymbol{T}}_a). \quad (11)$$

### 3.4 TRAINING OBJECTIVES

The goal of Atomas is to align the molecule and text at different levels of granularity while conditionally reconstructing the molecule or text description. We jointly optimize the global alignment loss $\mathcal{L}_{ga}$, hierarchical adaptive alignment loss $\mathcal{L}_{haa}$, and language modeling loss $\mathcal{L}_{lm}$ in an end-to-end manner. The overall loss function of Atomas simultaneously:

$$\min_\theta \mathcal{L}_{ga} + \mathcal{L}_{haa} + \mathcal{L}_{lm}, \quad \mathcal{L}_{haa} = \mathcal{L}_{haa}^a + \mathcal{L}_{haa}^f + \mathcal{L}_{haa}^m, \quad (12)$$

where $\theta$ denotes all learnable parameters of Atomas, $\mathcal{L}_{haa}^a$, $\mathcal{L}_{haa}^f$, $\mathcal{L}_{haa}^m$ operates at the atom level, fragment level, and molecule level, respectively.

## 4 EXPERIMENTS

In this section, we present the quantitative and qualitative results of Atomas. The experiment is set to evaluate the effectiveness of Atomas in four aspects: (1) improving the efficiency of **retrieval and property prediction tasks**, (2) enhancing the generation capability of **generation tasks**, (3) evaluating the **effectiveness of each module**, (4) chemical **significance and scalability**.

### 4.1 INITIAL TRAINING

**Dataset and Training Details:** We follow the MoleculeSTM's pipeline (Liu et al., 2023c) to collect molecular SMILES-text pairs from PubChem website. Pairs with the same PubChem ID and descriptions shorter than 18 characters are merged, and duplicates are removed from the downstream task datasets to prevent data leakage. This process results in a highquality dataset of 51,340 unique pairs, which is used for the initial training phase. Dataset details and statistics

Table 1: **Performance comparison on molecule-text retrieval task.** **Bold** and underlined indicate the best and second-best results, respectively. Details are provided in Section 4.2 and Appendix G.

| Model (No Fine-tuning) | Text to Molecule | | | | Molecule to Text | | | |
|---|---|---|---|---|---|---|---|---|
| | R@1 | R@5 | R@10 | MRR | R@1 | R@5 | R@10 | MRR |
| *1D SMILES + 2D Graph* | | | | | | | | |
| MoMu | 4.90 | 14.48 | 20.69 | 10.33 | 5.08 | 12.82 | 18.93 | 9.89 |
| MolCA | 35.09 | 62.14 | 69.77 | 47.33 | 37.95 | 66.81 | 74.48 | 50.80 |
| *1D SMILES + 2D Graph + Knowledge Graph* | | | | | | | | |
| MolFM | 16.14 | 30.67 | 39.54 | 23.63 | 13.90 | 28.69 | 36.21 | 21.42 |
| *1D SMILES* | | | | | | | | |
| MoleculeSTM | 35.80 | - | - | - | 39.50 | - | - | - |
| **Atomas-base (Ours)** | 39.08 | 59.72 | 66.56 | 47.33 | 37.88 | 59.22 | 65.56 | 47.81 |
| **Atomas-large (Ours)** | **49.08** | **68.32** | **73.16** | **57.79** | **46.22** | **66.02** | **72.32** | **55.52** |

are provided in Appendix E. The model is trained on 8 NVIDIA Tesla A100-SXM4-40GB GPUs using a batch size of 16 SMILES-text pairs, no weight decay, and the learning rate is set to $1e-4$. More implementation details are provided in Appendix F.

Table 2: **Performance comparison on text-based de novo molecule generation. Bold** and underlined indicate the best and second-best results, respectively. "↑" denotes that higher is better. "↓" denotes that lower is better. We repeat the Atomas-large 3 times and report the average with a 95% confidence interval. Details are provided in Section 4.3 and Appendix G.

| Model | BLEU↑ | Exact↑ | Levenshtein↓ | MACCS FTS↑ | RDK FTS↑ | Morgan FTS↑ | Validity↑ |
|---|---|---|---|---|---|---|---|
| *1D SMILES + 2D Graph + 2D Image* | | | | | | | |
| GIT-Mol | 0.756 | 0.051 | 26.32 | 0.738 | 0.582 | 0.519 | 0.928 |
| *1D SMILES + 2D Graph + Knowledge Graph* | | | | | | | |
| MolFM-small | 0.803 | 0.169 | 20.868 | 0.834 | 0.721 | 0.662 | 0.859 |
| MolFM-base | 0.822 | 0.210 | 19.45 | 0.854 | 0.758 | 0.758 | 0.892 |
| *2D Graph* | | | | | | | |
| ICMA(Galactica-125M)$_{4,2048}$ | 0.836 | - | 21.480 | 0.893 | 0.809 | 0.743 | 0.825 |
| ICMA(Mistral-7B)$_{4,2048}$ | 0.855 | - | 18.73 | 0.916 | 0.837 | 0.789 | 0.891 |
| *1D SMILES* | | | | | | | |
| MolT5-small | 0.749 | 0.082 | 28.816 | 0.780 | 0.654 | 0.601 | 0.725 |
| MolT5-base | 0.779 | 0.082 | 25.19 | 0.788 | 0.662 | 0.602 | 0.787 |
| MolT5-large | 0.854 | 0.318 | 16.32 | 0.889 | 0.813 | 0.750 | 0.958 |
| Text+Chem T5-augm | 0.853 | 0.322 | 16.87 | 0.901 | 0.816 | 0.757 | 0.943 |
| MolXPT | - | 0.215 | - | 0.859 | 0.757 | 0.667 | 0.983 |
| MolReGPT (GPT-3.5-turbo) | 0.790 | 0.139 | 24.91 | 0.847 | 0.708 | 0.624 | 0.887 |
| MolReGPT (GPT-4-0413) | 0.857 | 0.280 | 17.14 | 0.903 | 0.805 | 0.739 | 0.899 |
| **Atomas-base (Ours)** | 0.868 | 0.343 | 13.76 | 0.908 | 0.827 | 0.773 | 0.971 |
| **Atomas-large (Ours)** | **0.874**$^{\pm.003}$ | **0.387**$^{\pm.008}$ | **12.70**$^{\pm.28}$ | **0.914**$^{\pm.004}$ | **0.841**$^{\pm.002}$ | **0.788**$^{\pm.002}$ | **0.980**$^{\pm.003}$ |

Table 3: **Performance comparison on molecule captioning task. Bold** and underlined indicate the best and second-best results, respectively. We repeat the Atomas 3 times and report the average with a 95% confidence interval. Details are provided in Section 4.4 and Appendix G.

| Model | #Params | BLEU-2 | BLEU-4 | ROUGE-1 | ROUGE-2 | ROUGE-L |
|---|---|---|---|---|---|---|
| *1D SMILES + 2D Graph* | | | | | | |
| MoMu-small | 82M | 0.532 | 0.445 | - | - | 0.564 |
| MoMu-base | 252M | 0.549 | 0.462 | - | - | 0.575 |
| MoMu-large | 782M | 0.599 | 0.515 | - | - | 0.593 |
| InstructMol-GS | 6.9B | 0.475 | 0.371 | 0.566 | 0.394 | 0.502 |
| MolCA, Galac1.3B | 1.3B | 0.620 | 0.531 | 0.681 | 0.537 | 0.618 |
| *1D SMILES + 2D Graph + Image* | | | | | | |
| GIT-Mol-GS | 700M | 0.352 | 0.263 | 0.575 | 0.485 | 0.560 |
| *1D SMILES + 2D Graph + Knowledge Graph* | | | | | | |
| MolFM-small | 136.2M | 0.542 | 0.452 | 0.623 | 0.469 | 0.562 |
| MolFM-base | 296.2M | 0.585 | 0.498 | 0.653 | 0.508 | 0.594 |
| *1D SMILES* | | | | | | |
| MolT5-small | 77M | 0.519 | 0.436 | 0.620 | 0.469 | 0.563 |
| MolT5-base | 248M | 0.540 | 0.457 | 0.634 | 0.485 | 0.578 |
| MolT5-large | 783M | 0.594 | 0.508 | 0.654 | 0.510 | 0.594 |
| Text+Chem T5-augm | 220M | 0.625 | 0.542 | 0.682 | 0.543 | 0.622 |
| MolXPT | 350M | 0.594 | 0.505 | 0.660 | 0.511 | 0.597 |
| MolReGPT (GPT-3.5-turbo) | >175B | 0.565 | 0.482 | 0.450 | 0.543 | 0.585 |
| MolReGPT (GPT-4-0314) | - | 0.607 | 0.525 | 0.634 | 0.476 | 0.562 |
| **Atomas-base w/o initial training (Ours)** | 271M | 0.6045$^{\pm.003}$ | 0.5185$^{\pm.004}$ | 0.6745$^{\pm.006}$ | 0.5315$^{\pm.007}$ | 0.6155$^{\pm.004}$ |
| **Atomas-base (Ours)** | 271M | **0.632**$^{\pm.005}$ | **0.549**$^{\pm.002}$ | **0.685**$^{\pm.003}$ | **0.545**$^{\pm.004}$ | **0.626**$^{\pm.003}$ |

## 4.2 MOLECULE-TEXT RETRIEVAL

**Set Up:** To evaluate Atomas's performance and generalization, we use the PCdes dataset (Zeng et al., 2022b) instead of PubChem dataset, which includes 15,000 molecule pairs. Following MolFM (Luo et al., 2023), we apply scaffold splitting to divide the dataset into training, validation, and test sets at a 7:1:2 ratio. We directly evaluate Atomas and other baseline models on test sets without fine-tuning. For inference, we retrieve results directly from the entire test dataset without first selecting a top-k candidate set. We assess performance using Mean Reciprocal Rank (MRR) and Recall at 1, 5, and 10.

**Results:** Table 1 demonstrates that Atomas outperforms recent state-of-the-art methods in both text-to-molecule and molecule-to-text retrieval tasks on R@1. This indicates that multi-level fine-grained interaction and alignment can yield significantly better outcomes than methods based only on coarse-grained representations. For a detailed introduction to the baselines, refer to Appendix G.2.

## 4.3 TEXT-BASED DE NOVO MOLECULE GENERATION

**Set Up:** ChEBI-20 (Edwards et al., 2022) is a gold standard dataset extensively used for molecular generation tasks. It comprises 33,010 molecule-description pairs and is split into 80/10/10% train/validation/test sets. To assess our model's performance, we use standard metrics for the generation task. More details are provided in Appendices E and G.1.

Table 4: **Performance comparison on molecule property prediction.** We present the ROC-AUC (%) scores of molecular property prediction task on MoleculeNet. We use scaffold split following MoleculeSTM. We repeat the Atomas 3 times and report the average with a 95% confidence interval.

| Method | BBBP | Tox21 | ToxCast | Sider | ClinTox | MUV | HIV | Bace | Avg |
|---|---|---|---|---|---|---|---|---|---|
| MoleculeSTM-SMILES | 70.75±1.90 | 75.71±0.89 | 65.17±0.37 | 63.70±0.81 | 86.60±2.28 | 65.69±1.46 | 77.02±0.44 | 81.99±0.41 | 73.33 |
| MolFM | 72.9±0.1 | 77.2±0.7 | 64.4±0.2 | 64.2±0.9 | 79.7±1.6 | 76.0±0.8 | 78.8±1.1 | **83.9±1.1** | 74.62 |
| MoMu | 70.5±2.0 | 75.6±0.3 | 63.4±0.5 | 60.5±0.9 | 79.9±4.1 | 70.5±1.4 | 75.9±0.8 | 76.7±2.1 | 71.63 |
| MolCA-SMILES | 70.8±0.6 | 76.0±0.5 | 56.2±0.7 | 61.1±1.2 | 89.0±1.7 | - | - | 79.3±0.8 | 72.1 |
| **Atomas** | **73.72±1.67** | **77.88±0.36** | **66.94±0.9** | 64.40±1.9 | **93.16±0.5** | 76.30±0.7 | 80.55±0.43 | 83.14±1.71 | **77.01** |

Table 5: **The scaling of the initial training dataset on molecule generation task.** Atomas consistently surpasses baselines in limited data size.

| Model | Data sizes | Exact↑ | Levenshtein↓ | RDK FTS↑ |
|---|---|---|---|---|
| MolFM-base | 15k | 0.210 | 19.45 | 0.758 |
| Atomas-base | 0 | 0.298 | 15.47 | 0.809 |
| Atomas-base | 15k | 0.318 | 14.68 | 0.817 |
| **Atomas-base** | **51k** | **0.343** | **13.76** | **0.827** |

Table 6: **The scaling of the model size on molecule generation task.** Increasing Atomas parameters can enhance generation performance. Complete indicators are in the Appendix H.1.

| Model | Model sizes | Exact↑ | Levenshtein↓ | RDK FTS↑ |
|---|---|---|---|---|
| MolReGPT(GPT-4-0413) | >175B | 0.280 | 17.14 | 0.805 |
| MolT5-large | 783M | 0.318 | 16.32 | 0.813 |
| Atomas-base | 271M | 0.343 | 13.76 | 0.827 |
| **Atomas-large** | **825M** | **0.387** | **12.70** | **0.841** |

**Quantitative Results:** Tables 2, 9 and 14 show the text-based de novo molecule generation performance. Atomas outperforms all baseline models in all metrics. We also calculate the scores of molecule novelty to show that Atomas can perform well in generation. Since SMILES is the dominant molecular representation, Atomas uses SMILES and compares only with methods based on SMILES. Methods like (Liu et al., 2023a; Luo et al., 2023) use separate unimodal pre-trained models for different modalities, complicating information exchange and limiting fine-grained interactions. Their two-stage training process also restricts generative capabilities. Additionally, GPT-like and encoder-decoder-based methods (Edwards et al., 2022; Christofidellis et al., 2023; Liu et al., 2023d; Li et al., 2023a) miss the benefits of well-aligned multimodal representations.

## 4.4 MOLECULE CAPTIONING

**Set Up:** We evaluate Atomas for molecule generation using the ChEBI-20 dataset. Evaluation metrics include BLEU-2, BLEU-4, ROUGE-1, ROUGE-2, and ROUGE-L. We present more details in Appendices E and G.1.

**Quantitative Results:** Table 3 presents the overall molecule captioning performance. Atomas surpasses all baseline methods across all evaluation metrics. Notably, our Atomas-base model outperforms the MolT5-large model while using only 35.0% of its parameters and requiring no initial training, highlighting the effectiveness of our proposed framework.

## 4.5 MOLECULAR PROPERTY PREDICTION

**Set Up and Results:** We evaluate Atomas on eight binary classification datasets from MoleculeNet. We use scaffold split following (Liu et al., 2023c). The evaluation metric is the area under the receiver operating characteristic curve (ROC-AUC). Table 4 shows that we have consistent improvements on seven out of eight tasks compared to the baseline models using the SMILES string as input. The overall performance exceeds that of all baseline models.

## 4.6 ABLATION STUDY

**The Scalability and Robustness of Atomas:** Table 5 and Table 6 show that Atomas consistently outperforms baseline methods in both the scaling of training dataset and the scaling of model size. We also explore how Atomas's performance varies with complex molecular structures and textual descriptions. In the ChEBI-20 test dataset, molecules are categorized into length intervals of 100, with "Mol_len 100" representing lengths between 100 and 200. Similarly, the input molecule descriptions are categorized by length, with "Text_len 100" indicating descriptions between 100 and 200 characters. As shown in Tables 15 and 16, Atomas performs more robust performance than baseline methods on complex molecular structures and highly technical textual descriptions.

**Unified Encoder Better than Separate Encoders:** To investigate the impact of using a unified encoder versus two separate encoders for text and SMILES, we sample 75%, 50%, and 25% of

Table 7: **Ablation study for the effectiveness and time consumption of components on molecule generation task.** The first row is the baseline without alignment to generate SMILES based on text. The second row is the baseline only using global alignment. More details are in the Appendix H.3

| Global Alignmnet $\mathcal{L}_{ga}$ | Hierarchical Alignment $\mathcal{L}_{haa}$ | Conditional Generation $\mathcal{L}_{lm}$ | Exact↑ | Levenshtein↓ | Morgan FTS↑ | Training Time(s/sample) |
|---|---|---|---|---|---|---|
| | | ✓ | 0.082 | 24.846 | 0.602 | 0.0112 |
| ✓ | | ✓ | 0.223 | 16.946 | 0.716 | 0.0119 |
| | ✓ | ✓ | 0.266 | 16.675 | 0.736 | 0.0132 |
| ✓ | ✓ | ✓ | **0.298** | **15.472** | **0.750** | 0.0145 |

Table 8: **Ablation study for the effectiveness of joint optimization on molecule retrieval task.**

| Training Strategy | Text to Molecule | | | |
|---|---|---|---|---|
| | R@1 | R@5 | R@10 | MRR |
| 2Stages | 37.74 | 58.01 | 65.02 | 47.20 |
| **Joint optimization** | **39.08** | **59.72** | **66.56** | **48.47** |

| Training Strategy | Molecule to Text | | | |
|---|---|---|---|---|
| | R@1 | R@5 | R@10 | MRR |
| 2Stages | 36.54 | 57.31 | 63.58 | 46.10 |
| **Joint optimization** | **37.88** | **59.22** | **65.56** | **47.81** |

Table 9: **The scores of molecule novelty on text-based de novo molecule generation task.** And **The human expert evaluation on molecule caption task.** The numbers in brackets indicate the number of ranks 1, 2, and 3. The lower rank score indicates better performance of molecule caption.

| Method | Novelty↑ | Average Ranking of Human Expert Evaluation ↓ |
|---|---|---|
| Text+Chem T5-augm | 0.84 | 2.2(1/2/2) |
| MolT5-large | 0.76 | 2.6(0/2/3) |
| **Atomas-large** | **0.85** | **1.2(4/1/0)** |

the training set as training subsets, based on the distribution of text lengths, and evaluate on the original validation and test sets. As shown in Figure 3, the performance of the separate encoders (Sep-Encoder) significantly declines as the training sets decrease, compared to the unified encoder (Uni-Encoder). These findings may provide insight into molecular design, where data scarcity is a common challenge. Complete performance details are provided in Appendix H.2.

**The Effectiveness of Hierarchical Alignment:** The scarcity of fine-grained molecule and text pair datasets makes it challenging to quantify the model's ability to capture fine-grained information, unlike the vision community, where extensive datasets exist to evaluate models on fine-grained image recognition tasks. Nonetheless, we can implicitly validate the effectiveness of Atomas by observing improvements in component ablation studies. Table 7 provides clear insights into the effectiveness of fine-grained alignment. Figure 4 (right) shows the effect of adaptive alignment level numbers. We find that the model achieves the best performance at 3-level numbers.

**The Computational Efficiency:** Table 7 reports the average training time on the ChEBI-20 dataset using an NVIDIA A100 40 GB GPU and the performance on the molecule generation task. While hierarchical alignment slightly increases training time by just 0.0026 seconds, it significantly boosts Atomas's overall performance. The increase in inference time is almost negligible. This result highlights the effectiveness of our efficient design.

**Joint Optimization Benefits Both Learned Representation Quality and Generation Task Performance:** Table 8 and Figure 4 (left) show the ablation study on molecule retrieval and generation task for the different training strategies. The "Baseline" refers to the MolT5-base model. From these results, we can conclude that the essence of joint optimization is the mutual facilitation between the caption/generation task and molecular representation learning (retrieval tasks) (Bahdanau et al., 2015). More detailed discussions are provided in Appendix H.6.

**Merging Methods Comparison:** We compared Atomas with other molecular decomposition methods like BRICS decomposition method. The decomposition by BRICS resulted in fragments like'[1*]C(=O)C[7*]', which introduces additional characters and does not allow for hierarchical decomposition, limiting its direct replacement of the Adaptive Polymerization Module in Atomas.

## 4.7 VISUALIZATION AND QUALITATIVE ANALYSIS

**Visualization:** To better understand Atomas, we present the visualization of the adaptive polymerization module in Figure 5. The molecule is formed by combining atoms at positions 0-15 and at sites 16-26 through dehydration condensation. From atom level to fragment level, Atomas clusters atoms (words) into functional groups (phrases) using an adaptive polymerization module. From fragment level to molecule level, Atomas clusters atoms at sites 0-13 and 15 together to form a monomer-like structure. This indicates Atomas tends to focus on macro-level information as it ascends the hierarchy.

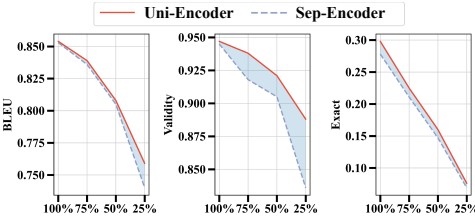

Figure 3: **Unified encoder vs separate encoder with the scaling dataset.** Evaluate on molecule generation task.

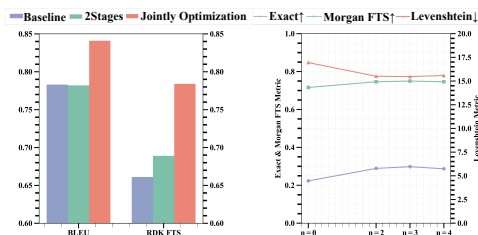

Figure 4: **Ablation study for the effectiveness of joint optimization** (left) and **hierarchical alignment level numbers** (right).

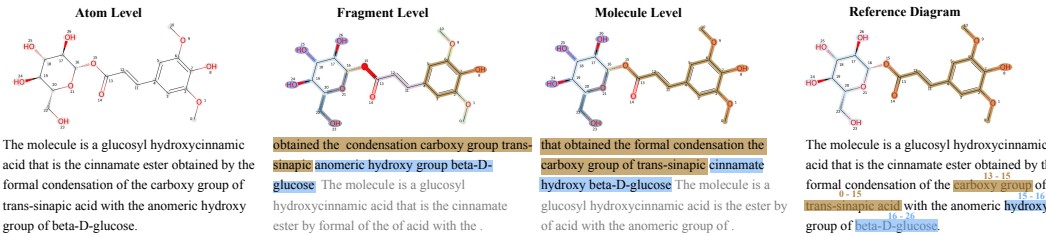

Figure 5: **The visualization of adaptive polymerization module.** The process of atom (word) polymerization to form individual sets is illustrated at three levels, including the reference diagram, from left to right. Atoms (words) belonging to the same set are highlighted using the same color.

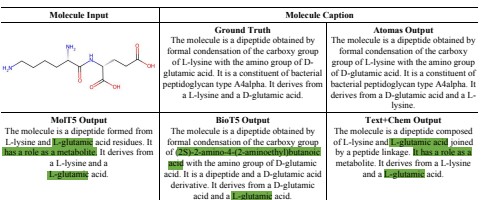

Figure 6: **Comparing the performance of global alignment methods with Atomas on molecule captioning task.**

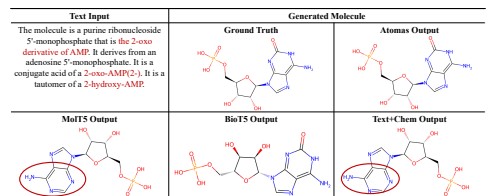

Figure 7: **Comparing the performance of global alignment methods with Atomas on molecule generation task.**

**Qualitative Analysis of Molecule Caption:** As shown in Figure 6, global alignment methods like (Christofidellis et al., 2023; Edwards et al., 2022; Pei et al., 2023) struggle to distinguish between "D-glutamate" and "L-glutamate" enantiomers. In contrast, Atomas generates more accurate and detailed molecule descriptions, demonstrating the effectiveness of hierarchical alignment models.

**Qualitative Analysis of Molecule Generation:** As shown in Figure 7. gloabl alignment methods like (Christofidellis et al., 2023; Edwards et al., 2022) can generate the "AMP" structure but miss fine-grained details like "2-hydroxy". Conversely, Atomas successfully generates the right structure.

**Human Evaluation:** To provide additional validation of the model's performance and practical utility, we incorporated human expert evaluations. We randomly selected five molecular captions of varying lengths generated by Atomas, Text+Chem T5, and MolT5. The average rankings from these evaluations are shown in Table 9. Atomas achieved the overall best performance among the three models, ranking first in 4 out of 5 generated molecular captions. The corresponding SMILES samples are provided in Appendix I.

## 5 CONCLUSION

We introduce Atomas, a hierarchical alignment framework designed to enhance molecular representation by aligning SMILES strings with textual descriptions through a Hierarchical Adaptive Alignment model. Atomas excels at capturing fine-grained details, achieving state-of-the-artin retrieval, property prediction, and molecule generation tasks while demonstrating robust scalability and generalizability.

## 6 ACKNOWLEDGEMENTS

Yikun Zhang would like to express sincere gratitude to everyone who contributed to this work, providing support and guidance throughout the process. Their valuable insights and assistance were instrumental in bringing this research to completion and enabling its publication at ICLR.

## ETHICS STATEMENT

This paper does not involve crowdsourcing or research with human subjects. The proposed method poses no high risk for misuse. We cite the original paper that produced the code package or dataset. Below is the broader impact of our research:

- **For machine learning community:** In this study, we introduce a Hierarchical Adaptive Alignment model for automatically learning fine-grained information. This offers a novel approach to facilitate the fine-grained learning of extensive unlabeled datasets in diverse domains.

- **For the drug discovery community:** Atomas utilizes the uni-encoder to alleviate the problem of limited data in the specific domain of molecular studies. This approach offers a novel training methodology for the data-scarce molecular drug discovery domain. Atomas's end-to-end training method involves alignment followed by generation and demonstrates superior performance in molecular understanding and generation tasks. Furthermore, Atoms provides the way for high-precision controllable molecular generation research. The adaptive alignment module in Atomas presents an efficient solution for leveraging large-scale unlabeled biochemical texts. We hope that the cross-modal representations learned from Atomas can be applied to a variety of molecular downstream tasks such as virtual screening and molecular drug design.

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

# Appendix

## A    COMPARISON TO RELATED WORKS

The primary challenge in multi-modal representation learning is effectively leveraging information from various modalities to learn common representations shared between them, *i.e.*, aligning different modalities. Building on the success of multi-modal representations in the vision community, exemplified by models such as CLIP(Radford et al., 2021b) and BLIP(Li et al., 2022), these advancements have found widespread applications in the life sciences, including domains such as small molecules(Liu et al., 2023c), proteins(Yuan et al., 2024), and materials(Ock et al., 2024).

Existing multi-modal molecule alignment approaches can be broadly categorized into two types: internal modalities and external modalities. Internal molecule structure representations include 1D fingerprints and molecule strings (specifically, SMILES - Simplified Molecular Input Line Entry System)(Weininger, 1988), 2D topological graphs(Yuan et al., 2025), and 3D conformational structures(Li et al., 2024b). External functional descriptions encompass textual descriptions and biological knowledge graphs.

### A.1    TEXT GUIDED CONDITIONAL MOLECULE GENERATION

Text-based molecule generation models can be primarily categorized into two types. One type uses a decoder-only transformer architecture, such as MolXPT (Liu et al., 2023d). This is a GPT-like model that utilizes the GPT-$2_{medium}$ configuration, which has been pre-trained on SMILES sequences encapsulated by text. The second type employs an encoder-decoder transformer architecture. This type translates between text and molecule strings, and it can adapt to the text-conditional de novo generation. Models like MolT5 and Text+Chem T5 (Christofidellis et al., 2023) work by jointly encoding the molecule string and natural language. They then use the input description to generate a molecule string.

In Table 10, we provide a comprehensive overview of existing works on molecule-text alignment methods. We have identified two key distinctions from existing methods: (i) Current contrastive learning-based alignment methods primarily align global features, neglecting finer-grained modal interactions. Fine-grained alignment is important in tasks such as controlled molecule generation and molecular captioning, as it enables greater precision and accuracy. (ii) Existing end-to-end training methods use conditional generation to create molecules, without aligning the molecule modality and text modality. However, our experimental results demonstrate that performing alignment before conditional generation can significantly improve generation performance.

Table 10: **Comparison between Atomas and existing molecule-and-text cross-modal methods.**

| Model | Input | | | | Alignment | Task | | | Training Strage | |
|---|---|---|---|---|---|---|---|---|---|---|
| | Text | Molecule | | | | Molecule Retrieval | Molecule Generation | Molecule Caption | Multi-Stage | End-to-End |
| | | 1D | 2D | 3D | | | | | | |
| MoMu (Su et al., 2022) | ✓ | ✓ | ✓ | - | ✓ | ✓ | ✓ | ✓ | ✓ | - |
| KV-PLM (Zeng et al., 2022b) | ✓ | ✓ | - | - | - | ✓ | - | - | ✓ | - |
| InstructMol (Cao et al., 2023) | ✓ | ✓ | ✓ | - | - | - | - | ✓ | ✓ | - |
| MolCA (Liu et al., 2023e) | ✓ | ✓ | ✓ | - | ✓ | ✓ | - | ✓ | ✓ | - |
| GIT-Mol (Liu et al., 2024) | ✓ | ✓ | ✓ | - | ✓ | ✓ | ✓ | - | ✓ | - |
| MolFM (Luo et al., 2023) | ✓ | ✓ | ✓ | - | ✓ | ✓ | ✓ | ✓ | ✓ | - |
| MolT5 (Edwards et al., 2022) | ✓ | ✓ | - | - | - | - | ✓ | ✓ | - | ✓ |
| Text+Chem T5 (Christofidellis et al., 2023) | ✓ | ✓ | - | - | - | - | ✓ | ✓ | - | ✓ |
| MolXPT (Liu et al., 2023d) | ✓ | ✓ | - | - | - | - | ✓ | ✓ | ✓ | - |
| MolReGPT (Li et al., 2023a) | ✓ | ✓ | - | - | - | - | ✓ | ✓ | - | - |
| MoleculeSTM (Liu et al., 2023c) | ✓ | ✓ | ✓ | - | ✓ | ✓ | - | - | ✓ | - |
| BioT5 (Pei et al., 2023) | ✓ | ✓ | - | - | - | - | ✓ | ✓ | - | ✓ |
| 3D-MOLM (Li et al., 2024b) | ✓ | ✓ | - | ✓ | ✓ | ✓ | - | ✓ | ✓ | - |
| ICMA (Li et al., 2024a) | ✓ | - | ✓ | - | ✓ | ✓ | ✓ | ✓ | ✓ | - |
| **Atomas (Ours)** | ✓ | ✓ | - | - | ✓ | ✓ | ✓ | ✓ | - | ✓ |

### A.2    DIFFRENCE WITH MOLECULAR MODALITIES REPRESENTATION LEARNING METHODS

**Different alignment objectives:** Intra-molecular modality vs Extra-molecular modality. (Feng et al., 2023; Yu et al., 2024; Ji et al., 2022; Liu et al., 2022) focus on aligning intra-molecular modalities, i.e., 1D SMILES, 2D molecule graph, and 3D structure, which is different from the

text-based molecule representation learning. This alignment benefits from easily obtainable molecular datasets and pair-wise alignments, facilitated by tools like RDKit for SMILES-to-graph conversion. Conversely, Atomas addresses text-driven tasks, involving cross-modal learning between intra- and extra-molecular modalities. Challenges arise from the lack of expert fine-grained textual annotations for molecules and difficulty in constructing positive/negative pairs, as a text fragment may suit multiple molecule substructures. These challenges make Atomas' achievements in this field particularly noteworthy.

**Different segmentation objectives:** (Feng et al., 2023) utilizes established algorithms for the segmentation of SMILES and graph representations. In contrast, Atomas is the first to segment both text descriptions and SMILES.

**Different fine-grained alignment methods:** Explicit vs Automatic. Atomas uses Hierarchical Adaptive Alignment, eliminating the need for explicit labeling between text fragments and molecular substructures. The weighted alignment allows for a flexible representation of cross-modal relationships. This approach is particularly beneficial in scenarios where the alignment between text and SMILES is not direct or where the textual data is rich in contextual information that requires sophisticated modeling.

**Different training objectives:** Prediction task vs Understanding and Generation task (Feng et al., 2023; Yu et al., 2024; Ji et al., 2022; Liu et al., 2022) are tailored for prediction tasks, while Atomas optimizes aligned representations for both understanding and generative tasks.

Table 11: **Comparison between Atomas and existing molecular modalities representation learning methods.**

| Model | Input | | | | Segmentation | | Task | | | |
|---|---|---|---|---|---|---|---|---|---|---|
| | Text | Molecule | | | Automatic | Explicit | Prediction | Molecule Retrieval | Molecule Generation | Molecule Caption |
| | | 1D-SMILES | 2D-Graph | 3D-Structure | | | | | | |
| GraphMVP (Liu et al., 2022) | - | - | ✓ | ✓ | - | ✓ | ✓ | - | - | - |
| UniMAP (Feng et al., 2023) | - | ✓ | ✓ | ✓ | - | ✓ | ✓ | - | - | - |
| MOLEBLEND (Yu et al., 2024) | - | - | ✓ | ✓ | - | - | ✓ | - | - | - |
| ReLMole (Ji et al., 2022) | - | - | ✓ | - | - | ✓ | ✓ | - | - | - |
| **Atomas (Ours)** | ✓ | ✓ | - | - | ✓ | - | - | ✓ | ✓ | ✓ |

## B    PRELIMINARIES

### B.1    MOLECULE REPRESENTATION

The structure of a molecule can be represented as a 1D molecular string. Specifically, SMILES is utilized to convert a chemical's 3D structure into a string of symbols. For instance, the structure of a benzene ring can be represented as a SMILES string: *C1=CC=C(C=C1)*. The 1D molecular string and the 2D graph are informationally equivalent, as SMILES can be losslessly converted to a graph using chemical toolkits (*e.g.* RDKit). Furthermore, transformer-based encoder models exhibit less information loss compared to Graph Neural Networks (GNNs) (Ma et al., 2022), which suffer from over-smoothing problems, and GNNs cannot perform unified encoding with text modality, posing challenges for interaction between the two modalities. In this study, we choose the 1D SMILES string and textual description for molecule cross-modal representation learning.

### B.2    ENCODER-DECODER T5 LANGUAGE MODEL

T5 (Raffel et al., 2020) is an encoder-decoder transformer-based model. In the self-supervised pre-training stage, for the input sequence $X$, some words in the sequence are randomly chosen for corruption. Each consecutive span of corrupted tokens is masked by a sentinel token(*e.g.* $< x >, < y >$). Then the objective is to reconstruct the dropped-out spans:

$$\mathcal{L}_{mlm}(X;\theta) = \mathop{\mathbb{E}}_{x \sim X} \mathop{\mathbb{E}}_{\text{mask}} \sum_{i \in \text{mask}} \log p\left(x_i \mid x_{j \notin \text{mask}}; \theta\right).$$

We use a 12-layer T5 model as the backbone and initialized using MolT5 (Edwards et al., 2022) weights, which have been pre-trained on both the textual modality C4 dataset (Raffel et al., 2020) and the molecular modality ZINC dataset (Sterling & Irwin, 2015) in a self-supervised manner.

### B.3 DENSITY PEAKS CLUSTERING ALGORITHM

The Density Peaks Clustering Algorithm (DPC) (Rodriguez & Laio, 2014) is a granular computing model that determines the number of clusters and their respective centers in a dataset by identifying density peaks. The algorithm is based on two assumptions:

*Assumption 1.* The local density of cluster centers (density peaks) is greater than the local density of their surrounding neighbors.

*Assumption 2.* The distance between different cluster centers is relatively large. Briefly, given a dataset $D = \{x_1, x_2, \ldots, x_n\}$ with $n$ samples, the local density of $x_i$ is defined as $\rho_i = \sum_{j=1}^{n} \chi(d_{ij} - d_c)$, where $\chi$ is an indicator function: $\chi(x) = 1$ when $x < 0$, and $\chi(x) = 0$ otherwise, and $d_c$ a cutoff distance. The relative distance $\delta$ is defined as $\delta_i = \min_{j:\rho_j > \rho_i}(d_{ij})$. Based on $\rho_i$ and $\delta$, DPC algorithm constructs decision graphs to classify data points $x_i$ as density peak points, normal points, or outliers.

## C   LIMITAIONS

As Atomas demonstrated excellent generation and retrieval capabilities, there may concerns about potential overfitting. To validate the generalization ability of Atomas, we used the Out-of-Distribution (OOD) dataset, PCdes, in the retrieval task. Atomas outperformed recent state-of-the-art methods in both text-to-molecule and molecule-to-text retrieval tasks on R@1, indicating robust generalization.

## D   ALGORITHM

---

**Algorithm 1** The proposed hierarchical cross-modal molecular representation learning framework that jointly learns representations from SMILES and text.

---

1: **Input:** SMILES strings $S$, Text descriptions $T$
2: **Output:** Aligned representations for molecules and texts
3: **procedure** UNIFIEDENCODER($S$, $T$)
4:     $S_{embed} \leftarrow$ Encode($S$)
5:     $T_{embed} \leftarrow$ Encode($T$)
6:     **return** $S_{embed}, T_{embed}$
7: **end procedure**
8: **procedure** ADAPTIVEPOLYMERIZATION($S_{embed}, T_{embed}$)
9:     **for** each level in {atom, fragment, molecule} **do**
10:         $S_{clustered} \leftarrow$ ClusterTokens($S_{embed}$)
11:         $T_{clustered} \leftarrow$ ClusterTokens($T_{embed}$)
12:         $S_{embed}, T_{embed} \leftarrow$ MergeClusters($S_{clustered}, T_{clustered}$)
13:     **end for**
14:     **return** $S_{embed}, T_{embed}$
15: **end procedure**
16: **procedure** WEIGHTEDALIGNMENTMODULE($S_{embed}, T_{embed}$)
17:     **for** each level in {atom, fragment, molecule} **do**
18:         $alignment \leftarrow$ ComputeAlignment($S_{embed}, T_{embed}$)
19:         $S_{embed}, T_{embed} \leftarrow$ UpdateRepresentations($S_{embed}, T_{embed}, alignment$)
20:     **end for**
21:     **return** $S_{embed}, T_{embed}$
22: **end procedure**
23: **procedure** CONDITIONALDECODER($S_{embed}, T_{embed}$)
24:     $GeneratedOutput \leftarrow$ Decode($S_{embed}, T_{embed}$)
25:     **return** $GeneratedOutput$
26: **end procedure**
27: **Begin**
28: $S_{embed}, T_{embed} \leftarrow$ UNIFIEDENCODER($S, T$)
29: $S_{embed}, T_{embed} \leftarrow$ ADAPTIVEPOLYMERIZATION($S_{embed}, T_{embed}$)
30: $S_{embed}, T_{embed} \leftarrow$ WEIGHTEDALIGNMENTMODULE($S_{embed}, T_{embed}$)
31: $Output \leftarrow$ CONDITIONALDECODER($S_{embed}, T_{embed}$)
32: **End**

---

## E   DATA DETAILS

### E.1   INITIAL TRAINING DATASET CONSTRUCTION

We obtain a dataset of 280K molecule-text pairs from PubChem database and follow the MolecularSTM to preprocess the textual descriptions, named PubchemSTM-raw. Molecule names are replaced by "This molecule is ..."or "These molecules are ...."to prevent the model from identifying molecules by name alone. To create unique SMILES-text pairs, molecules with the same CID (Chemical Identifier) are combined, resulting in 243K pairs. Text descriptions with less than 18 characters are filtered out, resulting in a set of 64,285 samples. In order to avoid data leakage, we removed duplicates from the ChEBI-20 and PCdes datasets used in downstream tasks. Specifically, we first convert the smiles string into a canonical SMILES string using the RDKit toolkit, and then

de-duplicate the initial training dataset with the same SMILES string as the ChEBI-20 dataset and PCdes datasets, respectively. This resulted in a high-quality and leak-free dataset of 51,340 pairs, named PubchemSTM-distll. It should be noted that PubchemSTM-distll is used exclusively for initial training and does not divide the training, testing, and validation sets.

## E.2 DATASET STATISTIC

Table 12 presents the dataset statistics. We observe that PubchemSTM-raw includes many uninformative texts; for example, some descriptions include just one word, such as "4,4'-Methylenebis". and one molecule corresponds to multiple descriptions. So, we first create unique SMILES-text pairs by grouping pairs by CID, and then filter out the texts with a length of less than 18 characters.

Table 12: **Statistics of the datasets.**

| Dataset | Molecule-Text Pair | Train | Valid | Test | Min Word | Avg Word | Median Word |
|---|---|---|---|---|---|---|---|
| PubchemSTM-raw | 280011 | - | - | - | 1 | 18.37 | 13 |
| PubchemSTM-distll | 51340 | 51340 | 0 | 0 | 18 | 44.64 | 30 |
| ChEBI-20 | 33008 | 26407 | 3301 | 3300 | 18 | 43.49 | 40 |
| PCdes | 14995 | 10495 | 1500 | 3000 | 17 | 61.2 | 50 |

## E.3 DATASET EXAMPLES

Figure 8 shows some examples of our dataset. To make it easier to understand, we use the RDKit toolkit to convert SMILES strings into a 2D molecular graph. As shown in Figure 8, the text descriptions available to us contain detailed local descriptions of molecular structures. The corresponding parts of the text and molecular structures are highlighted with the same color in the table for clarity. Effectively utilizing these localized descriptions is crucial for enhancing the performance of text-based controlled molecule generation tasks. The result of our experiments offers substantial evidence supporting the significance of leveraging these fine-grained descriptions in the generation process.

| Molecule Graph | SMILES | Description |
|---|---|---|
|  | C1=CC(=CC=C1C2=CC(=O)C3=C(O2)C=C(C(=C3O)C4=C(C=C(C5=C4OC(=CC5=O)C6=CC=C(C=C6)O)O)O)O)O | The molecule is a biflavonoid that is obtained by oxidative coupling of two molecules of apigenin resulting in a bond between positions C-6 and C-8 of the two chromene rings. It has a role as an antineoplastic agent, an antiviral agent, a hepatoprotective agent and a metabolite. It is a biflavonoid, a hydroxyflavone and a biaryl. |
|  | C1=CC(=C(C=C1CC(C(=O)O)N)O)O | The molecule is a hydroxyphenylalanine carrying hydroxy substituents at positions 3 and 4 of the benzene ring. It has a role as a human metabolite. It is a hydroxyphenylalanine, a tyrosine derivative and a non-proteinogenic alpha-amino acid. |
|  | CC(=O)N1CCC(CC1)NC(=O)NC2=CC=C(C=C2)OC(F)(F)F | The molecule is a phenylurea that is urea substituted by 1-acetylpiperidin-4-yl and 4-(trifluoromethoxy)phenyl groups at positions 1 and 3 respectively. It has a role as an EC 3.3.2.10 (soluble epoxide hydrolase) inhibitor. |

Figure 8: **Examples of dataset.**

# F INITIAL TRAINING DETAILS

## F.1 TRAINING SET UP

For the backbone models, we choose MolT5-base which is pre-trained in masked language modeling manner on two uni-modal datasets: natural language dataset and molecular dataset. For the model implementation, we use the PyTorch Lightning framework and employ distributed parallel training. We use the AdamW optimizer with no weight decay and the learning rate is set to $1e^{-4}$. The key hyperparameters used in Atomas are illustrated in the table Table 13.

Table 13: **Hyperparameter details for Atomas.**

| Hyperparameter | Value |
|---|---|
| batchsize | 16 |
| epoch | 100 |
| encoder learning rate | 1e-4 |
| decoder learning rate | 1e-4 |
| text projection learning rate | 1e-5 |
| molecule projection learning rate | 1e-5 |
| max padding length | 512 |
| queue size | 13200 |
| precision | BFloat16 Automatic Mixed Precision |

## F.2 TRAINING ANALYSIS

We visualized the loss curve for training 100 epochs during initial training. We scaled the loss value using a logarithmic scale. Figure 9a shows the loss curve of the global alignment $\mathcal{L}_{ga}$ after the addition of hierarchical adaptive alignment loss $\mathcal{L}_{haa}$. Figure 9b shows the loss curve of language modeling $\mathcal{L}_{lm}$ after the addition of the hierarchical adaptive alignment loss $\mathcal{L}_{haa}$. The observations suggest that the hierarchical adaptive alignment enhances global alignment and controllable generation.

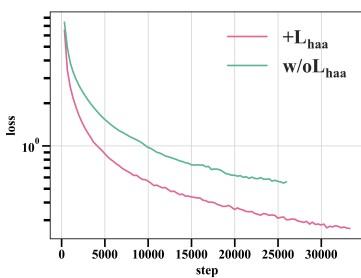
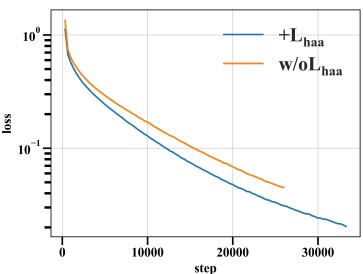

(a) The convergence of $\mathcal{L}_{ga}$ loss both in the absence and presence of $\mathcal{L}_{haa}$ loss.

(b) The convergence of $\mathcal{L}_{lm}$ loss both in the absence and presence of $\mathcal{L}_{haa}$ loss

## G    DOWNSTREAM TASK DETAILS

### G.1    METRICS

**Metrics of Molecule-Text Retrieval:** In retrieval task, we are consistent with the natural scene, using the most common search indicators in the natural scene (*i.e.*, text-image retrieval (Radford et al., 2021a)). Recall at 1/5/10 (Manning et al., 2008) is a performance metric for information retrieval systems, such as search engines or recommendation systems, that measures the proportion of relevant results found within the top 1, 5, or 10 returned items, indicating the model's effectiveness in retrieving pertinent information. MRR(Voorhees, 2000): mean reversed rank. MRR evaluates information retrieval model by averaging the inverse positions of the first relevant results across multiple queries, reflecting the model's effectiveness in ranking relevant items.

**Metrics of Molecule Captioning:** In molecule catpioning task, we follow the MolT5(Edwards et al., 2022) model and employ BLEU (Bilingual Evaluation Understudy) (Papineni et al., 2002) and ROUGE (Recall-Oriented Understudy for Gisting Evaluation) (Lin, 2004) as evaluation metrics for captions. BLEU measures the overlap of n-grams between the generated text and the reference text. BLEU-n refers to the BLEU metric with n-grams, where n is an integer value (*e.g.*, 1 for unigrams, 2 for bigrams, 3 for trigrams). For example, BLEU-1 measures the accuracy of word level, and higher-order BLEU can measure the fluency of sentences. The BLEU score ranges between 0 and 1. A BLEU score of 0.6 or 0.7 is considered to be a good result. ROUGE-N measures the overlap of N-grams (*e.g.*, unigrams, bigrams, trigrams) between the generated and reference texts. It calculates precision, recall, and F-score for N-grams, providing a balanced assessment of the model's performance. ROUGE-L is based on the longest common subsequence (LCS) between the generated and reference texts. It considers the longest continuous sequence of words that appear in both texts, capturing the overall coherence and flow of the generated text.

**Metrics of Text-based de Novo Molecule Generation:** In molecule generation task we use BLEU, Exact, Levenshtein (Miller et al., 2009), MACCS FTS (Durant et al., 2002), RDK FTS (Schneider et al., 2015), Morgan FTS (Rogers & Hahn, 2010), and Validity 7 metric. Exact refers to the Exact Match metric measures the percentage of predictions that exactly match the true labels. The Levenshtein distance, also known as the edit distance, is a metric used to measure the similarity between two strings by calculating the minimum number of single-character edits (insertions, deletions, or substitutions) required to transform one string into the other. Validity refers to the percentage of molecules that can be processed by the RDKIT and measures the grammatical normality of the generated molecules. MACCS FTS, RDK FTS and Morgan FTS are molecule fingerprint metrics.

### G.2    BASELINES

**Baselines of Molecule-Text Retrieval:** MolFM (Luo et al., 2023) jointly trains three unimodal encoders, which separately encode molecular structures, biomedical texts, and knowledge graphs to learn joint representations. MoMu (Su et al., 2022) is a pre-trained model that utilizes contrastive learning to align molecular graphs with their corresponding text descriptions. MoleculeSTM (Liu et al., 2023c) designs a multi-modal contrastive learning model for molecular understanding that incorporates both molecular structural information and textual knowledge. MolCA (Liu et al., 2023e) enables language models to understand both text- and graph-based molecular contents via the cross-modal projector.

**Baselines of Molecule Captioning:** We compared 9 baselines including MoMu, MolXPT, GIT-Mol (Liu et al., 2024), MolFM, MolT5, MolReGPT, Text+Chem T5, InstructMol (Cao et al., 2023) and MolCA (Liu et al., 2023e). MolXPT is a GPT-like model that uses the GPT-2$_{medium}$ configuration pre-trained on SMILES sequences wrapped by text. GIT-Mol maps molecular graphs, images, and text SMILES modalities into a unified latent space using a GIT-Former designed based on the Q-Former architecture in BLIP2 (Li et al., 2023b). MolT5 is a T5-based text-to-text model, pre-trained on a large-scale single-modal corpus of natural language and molecules, thereby obtaining prior knowledge of the two domains. MolReGPT employs GPT-3.5-turbo and GPT-4-0314, and designs a retrieval-based prompt paradigm through in-context learning to improve molecule discovery without any additional training. Text+Chem T5 develops a multi-task, multi-domain model for natural and chemical language. InstructMol employs instruction-tuning manner through two-stage training to fine-tune LLMs (large language models). InstructMol+GS refers to the use of both molecular graph

tokens and SMILES tokens as input. MolCA bridges molecular 2D graph and text by projecting the graph into a semantic space similar to text.

**Baselines of Text-based de Novo Molecule Generation:** We compared the performance of six out of seven baselines used in molecule generation task. We excluded the InstructMol method from the comparison, as it is not directly applicable to the molecule generation task.

## G.3 SUPPLEMENTARY EXPERIMENT FOR TEXT-BASED DE NOVO MOLECULE GENERATION TASK ON PUBCHEM324K DATASET

We conducted an additional experiment to compare Atomas' performance with the state-of-the-art model, ICMA (Li et al., 2024a), on the PubChem dataset provided by MolCA (Liu et al., 2023e). Specifically, we trained Atomas-large using only the PubChem324K training dataset, aligning the experimental settings with ICMA by ignoring pretaining data. The results are presented in the . Based on the data presented, Atomas achieves significant improvements on most metrics over ICMA, including a 63.11% improvement on the BLEU metric and a 46.54% improvement on Levenshtein compared to ICMA. This demonstrates Atomas' superior performance on the PubChem dataset, highlighting its robust generalization capabilities.

Table 14: **Performance comparison of text-based de novo molecule generation on PubChem324k test dataset. Bold** and underlined indicate the best and second-best results, respectively. "↑" denotes that higher is better. "↓" denotes that lower is better.

| Model | Model sizes | BLEU↑ | Levenshtein↓ | MACCS FTS↑ | RDK FTS↑ | Morgan FTS↑ | Validity↑ |
|---|---|---|---|---|---|---|---|
| ICMA(Galactica-125M)$_{4,2048}$ | 125M | 0.569 | 52.75 | 0.719 | 0.579 | **0.652** | 0.825 |
| ICMA(Mistral-7B)$_{4,2048}$ | 7B | 0.450 | 77.01 | 0.764 | 0.624 | 0.504 | 0.891 |
| **Atomas-large** | 825M | **0.734** | **28.186** | **0.773** | **0.637** | 0.535 | **0.945** |

# H ABLATION STUDY

## H.1 THE SCALABILITY AND ROBUSTNESS OF ATOMAS

Tables 15 and 16 show Atomas perform on complex molecular structures and textual descriptions. "Mol_len 100" and "Text_len 100" indicate lengths from 100 to 200. Atomas performs more robust performance than baseline methods on complex molecular structures and highly technical textual descriptions.

Table 15: **Performance comparison on complex structures using CHEBI20 test dataset on molecule caption task.**

| Mol-Len | Atomas-BLEU2 | Text+Chem T5-augm-BLEU2 | MolT5-BLEU2 | Atomas-ROUGE-1 | Text+Chem T5-augm-ROUGE-1 | MolT5-ROUGE-1 | Atomas-ROUGE-L | Text+Chem T5-augm-ROUGE-L | MolT5-ROUGE-L |
|---|---|---|---|---|---|---|---|---|---|
| 100 | 0.712 | 0.629 | 0.653 | 0.745 | 0.691 | 0.702 | 0.693 | 0.640 | 0.651 |
| 200 | 0.679 | 0.588 | 0.603 | 0.722 | 0.651 | 0.669 | 0.659 | 0.599 | 0.613 |
| 300 | 0.742 | 0.607 | 0.699 | 0.785 | 0.670 | 0.717 | 0.741 | 0.619 | 0.673 |
| 400 | 0.692 | 0.464 | 0.612 | 0.749 | 0.521 | 0.651 | 0.688 | 0.454 | 0.583 |
| 500 | 0.760 | 0.639 | 0.736 | 0.756 | 0.726 | 0.768 | 0.686 | 0.679 | 0.714 |
| 600 | 0.811 | 0.470 | 0.638 | 0.842 | 0.683 | 0.662 | 0.793 | 0.601 | 0.662 |
| 700 | 0.588 | 0.516 | 0.583 | 0.675 | 0.644 | 0.672 | 0.637 | 0.536 | 0.637 |
| 800 | 0.388 | 0.430 | 0.560 | 0.488 | 0.528 | 0.559 | 0.429 | 0.466 | 0.530 |
| 900 | 0.352 | 0.057 | 0.331 | 0.494 | 0.263 | 0.488 | 0.414 | 0.248 | 0.442 |

Table 16: **Performance comparison on complex textual descriptions using CHEBI20 test dataset on generation task.**

| Text-Len | Atomas-BLEU↑ | Text+Chem T5-augm-BLEU↑ | MolT5-BLEU↑ | Atomas-Levenshtein↓ | Text+Chem T5-augm-Levenshtein↓ | MolT5-Levenshtein↓ | Atomas-Morgan FTS↑ | Text+Chem T5-augm-Morgan FTS↑ | MolT5-Morgan FTS↑ |
|---|---|---|---|---|---|---|---|---|---|
| 100 | 0.867 | 0.783 | 0.842 | 10.218 | 17.140 | 13.084 | 0.902 | 0.849 | 0.864 |
| 200 | 0.888 | 0.849 | 0.871 | 10.234 | 16.455 | 13.777 | 0.930 | 0.901 | 0.907 |
| 300 | 0.883 | 0.867 | 0.861 | 13.236 | 17.619 | 16.137 | 0.921 | 0.901 | 0.908 |
| 400 | 0.861 | 0.793 | 0.832 | 17.463 | 28.942 | 22.668 | 0.910 | 0.882 | 0.881 |
| 500 | 0.858 | 0.828 | 0.845 | 22.127 | 34.035 | 26.479 | 0.874 | 0.831 | 0.845 |
| 600 | 0.793 | 0.807 | 0.800 | 35.789 | 40.281 | 38.912 | 0.817 | 0.805 | 0.794 |
| 700 | 0.667 | 0.642 | 0.640 | 58.789 | 65.579 | 61.632 | 0.817 | 0.786 | 0.818 |
| 800 | 0.878 | 0.845 | 0.820 | 36.857 | 42.429 | 39.143 | 0.929 | 0.803 | 00.790 |
| 900 | 0.809 | 0.549 | 0.528 | 14.001 | 24.030 | 25.200 | 0.963 | 0.672 | 0.550 |

Tables 17 and 18 presents the complete results of the two scaling experiments to show Atomas's scalability and robustness.

Table 17: **The scaling of the initial training dataset on molecule generation task** Here we choose 15k training data to be consistent with MolFM which is one of the baselines in the paper.

| Model | Data sizes | BLEU↑ | Exact↑ | Levenshtein↓ | MACCS FTS↑ | RDK FTS↑ | Morgan FTS↑ | Validity↑ |
|---|---|---|---|---|---|---|---|---|
| Atomas-base | 0 | 0.854 | 0.298 | 15.47 | 0.898 | 0.809 | 0.750 | 0.947 |
| Atomas-base | 15k | 0.861 | 0.318 | 14.68 | 0.902 | 0.817 | 0.757 | 0.965 |
| **Atomas-base** | 51k | **0.868** | **0.343** | **13.76** | **0.908** | **0.827** | **0.773** | **0.971** |

Table 18: **The scaling of the model size on molecule generation task** The results presented below demonstrate that increasing the parameters of Atomas can lead to further improvements generation performance.

| Model | Model sizes | BLEU↑ | Exact↑ | Levenshtein↓ | MACCS FTS↑ | RDK FTS↑ | Morgan FTS↑ | Validity↑ |
|---|---|---|---|---|---|---|---|---|
| MolReGPT (GPT-3.5-turbo) | >175B | 0.790 | 0.139 | 24.91 | 0.847 | 0.708 | 0.624 | 0.887 |
| MolReGPT (GPT-4-0413) | >175B | 0.857 | 0.280 | 17.14 | 0.903 | 0.805 | 0.739 | 0.899 |
| MolT5-base | 248M | 0.779 | 0.082 | 25.19 | 0.788 | 0.662 | 0.602 | 0.787 |
| MolT5-large | 783M | 0.854 | 0.318 | 16.32 | 0.889 | 0.813 | 0.750 | 0.958 |
| Atomas-base | 271M | 0.868 | 0.343 | 13.76 | 0.908 | 0.827 | 0.773 | 0.971 |
| **Atomas-large** | 825M | **0.874** | **0.387** | **12.70** | **0.914** | **0.841** | **0.788** | **0.980** |

## H.2 ABLATION STUDY FOR UNIFIED ENCODER VS SEPARATE ENCODER

Table 19 presents the advantages of utilizing a unified encoder in the Atomas.

Figures 10a to 10d illustrates the distribution of the ChEBI-20 training data after weighted sampling. These data is utilized to verify that the unified encoder outperforms separate encoders in scenarios with limited data availability.

Table 19: **Ablation study for the use of the unified encoder and separate encoder.** Performance on the text-based de novo molecule generation task using the ChEBI-20 dataset. "↑" denotes that higher is better. "↓" denotes that lower is better.

| Method | BLEU↑ | Exact↑ | Levenshtein↓ | MACCS FTS↑ | RDK FTS↑ | Morgan FTS↑ | Validity↑ |
|---|---|---|---|---|---|---|---|
| Baseline | 0.783 | 0.082 | 24.846 | 0.788 | 0.661 | 0.602 | 0.787 |
| Sep-encoder | 0.853 | 0.278 | 15.72 | 0.895 | 0.805 | 0.745 | 0.945 |
| **Uni-encoder** | **0.854** | **0.298** | **15.472** | **0.898** | **0.809** | **0.750** | **0.947** |

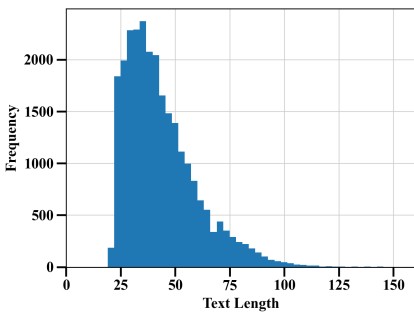

(a) The text length distribution of full ChEBI-20 training dataset.

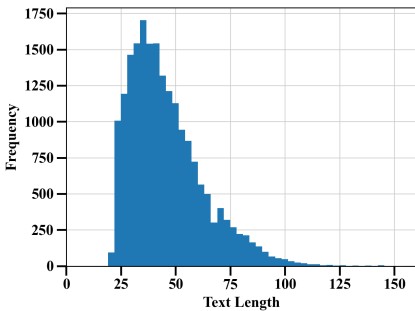

(b) The distribution of text length in 75% of the ChEBI-20 training dataset after applying weighted sampling.

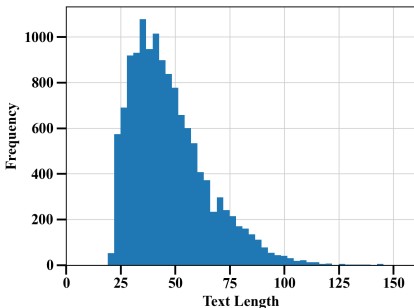

(c) The distribution of text length in 50% of the ChEBI-20 training dataset after applying weighted sampling.

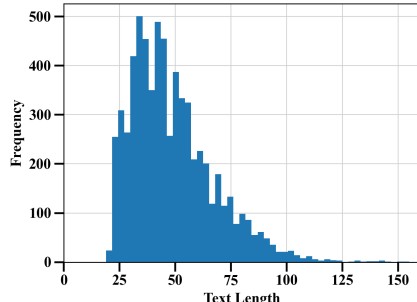

(d) The distribution of text length in 25% of the ChEBI-20 training dataset after applying weighted sampling.

## H.3    ABLATION STUDY FOR THE EFFECTIVENESS OF COMPONENTS.

Table 20 presents the complete results of the ablation study for the effectiveness of components.

Table 20: **Ablation study for the effectiveness of components.** Performance on the text-based de novo molecule generation task using the ChEBI-20 dataset. "↑" denotes that higher is better. "↓" denotes that lower is better.

| $\mathcal{L}_{ga}$ | $\mathcal{L}_{haa}$ | $\mathcal{L}_{lm}$ | BLEU↑ | Exact↑ | Levenshtein↓ | MACCS FTS↑ | RDK FTS↑ | Morgan FTS↑ | Validity↑ |
|---|---|---|---|---|---|---|---|---|---|
| | | ✓ | 0.783 | 0.082 | 24.846 | 0.788 | 0.661 | 0.602 | 0.787 |
| ✓ | | ✓ | 0.841 | 0.223 | 16.946 | 0.886 | 0.784 | 0.716 | **0.954** |
| | ✓ | ✓ | 0.844 | 0.266 | 16.675 | 0.893 | 0.799 | 0.736 | 0.952 |
| ✓ | ✓ | ✓ | **0.854** | **0.298** | **15.472** | **0.898** | **0.809** | **0.750** | 0.947 |

## H.4    ABLATION STUDY FOR THE EFFECTIVENESS OF JOINT OPTIMIZATION.

Table 21 presents the complete results of the ablation study for the effectiveness of joint optimization.

Table 21: **Ablation study for the effectiveness of joint optimization.** Performance on the text-based de novo molecule generation task using the ChEBI-20 dataset.

| Method | BLEU↑ | Exact↑ | Levenshtein↓ | MACCS FTS↑ | RDK FTS↑ | Morgan FTS↑ | Validity↑ |
|---|---|---|---|---|---|---|---|
| Baseline | 0.783 | 0.082 | 24.846 | 0.788 | 0.661 | 0.602 | 0.787 |
| 2Stages | 0.782 | 0.106 | 26.029 | 0.812 | 0.689 | 0.602 | 0.910 |
| **Jointly optimization** | **0.841** | **0.223** | **16.946** | **0.886** | **0.784** | **0.716** | **0.954** |

## H.5    ABLATION STUDY FOR THE EFFECTIVENESS OF DIFFERENT NUMBER OF HIERARCHICAL ALIGNMENT LEVELS.

Table 22 presents the complete results of the effect of different number of hierarchical alignment levels.

Table 22: **Ablation study for the effect of different number of hierarchical alignment levels.** Performance on the text-based de novo molecule generation task using the ChEBI-20 dataset.

| Level Num | BLEU↑ | Exact↑ | Levenshtein↓ | MACCS FTS↑ | RDK FTS↑ | Morgan FTS↑ | Validity↑ |
|---|---|---|---|---|---|---|---|
| 0 | 0.841 | 0.223 | 16.946 | 0.886 | 0.784 | 0.716 | **0.954** |
| 2 | 0.854 | 0.289 | 15.506 | 0.896 | 0.805 | 0.746 | 0.950 |
| **3** | **0.854** | **0.298** | **15.472** | **0.898** | **0.809** | **0.750** | 0.947 |
| 4 | 0.852 | 0.287 | 15.580 | 0.897 | 0.808 | 0.746 | 0.952 |

## H.6    JOINT OPTIMIZATION BENEFITS BOTH LEARNED REPRESENTATION QUALITY AND GENERATION TASK PERFORMANCE.

The essence of joint optimization is the mutual facilitation between the caption/generation task and molecular representation learning (retrieval tasks). Here we provide the in-depth discussion of this joint optimization strategy. *From model perspective:* As suggested by the existing study (Bahdanau et al., 2015), attention-based generation tasks essentially perform a form of soft alignment. During the generation process, the attention mechanism facilitates a mutual translation between text and SMILES, reinforcing the semantic consistency between the textual description and the molecular structure it represents. Concurrently, representation learning bridges the domain gap between text and SMILES, enhancing the caption/generation task. *From data perspective:* The captioning and generation tasks may provide complementary information for learning molecular representations. These tasks necessitate the model to learn the mapping between text and molecular domains, which allows the model to grasp the intricate relationship between textual descriptions and molecular structures, thereby enriching the quality of the learned molecular representations—hierarchical alignment further aids in capturing local data pair relationships, benefiting the generation process.

## I  HUMAN EVALUATION

To provide additional validation of the model's performance and practical utility, we incorporated human expert evaluations. We randomly selected five molecular captions of varying lengths generated by Atomas, Text+Chem T5, and MolT5. The average rankings from these evaluations are shown in Table 9, where a lower average ranking indicates better performance. Atomas achieved the overall best performance among the three models, ranking first in 4 out of 5 generated molecular captions.

Below are the SMIELS samples tested by human expert:

**SMILES(1):**  C[C@@]12CC[C@@H]3[C@@]([C@H]1C[C@H]([C@]4([C@H]2CCC5=C4C(=O)OC5)C)O)(CC[C@@H](C3(C)C)O)C

**SMILES(2):**  C[C@@H](C(=O)N[C@H](CCC(=O)NCCCC[C@H](C(=O)O)N)C(=O)O)NC(=O)[C@@H](C)O[C@@H]1[C@H]([C@H](O[C@@H]([C@H]1O)CO)OP(=O)(O)OP(=O)(O)OC[C@@H]2[C@H]([C@H]([C@@H](O2)N3C=CC(=O)NC3=O)O)O)NC(=O)C

**SMILES(3):**  C[C@@H]1[C@@H](C[C@H]([C@H](O1)O[C@H]2[C@@H]([C@H](O[C@@H]([C@H]2O[C@@H]3[C@@H]([C@H]([C@H]([C@H](O3)CO)O)O[C@H]4[C@@H]([C@@H]([C@H]([C@@H](O4)C)O)O)O)O)O[C@H]5[C@@H](O[C@H]([C@@H]([C@@H]5O)O)O)C)CO)O)O)O

**SMILES(4):**  C[C@H]1[C@@H]([C@@](C[C@@H](O1)O[C@@H]2[C@H]([C@@H]([C@H](O[C@H]2OC3=C4C=C5C=C3OC6=C(C=C(C=C6)[C@H]([C@H]7C(=O)N[C@@H](C8=C(C(=CC(=C8)O)O)C9=C(C=CC(=C9)[C@H](C(=O)N7)NC(=O)[C@@H]5NC(=O)[C@@H](NC(=O)[C@@H]([C@@H](C1=CC(=C(O4)C=C1)Cl)O)NC(=O)[C@@H](CC(C)C)[NH2+]C)CC(=O)N)O)C(=O)[O-])O[C@H]1C[C@]([C@H]([C@@H](O1)C)O)(C)[NH3+])Cl)CO)O)O)(C)[NH3+])O

**SMILES(5):**  C[C@H]1[C@H]([C@H]([C@@H]([C@@H](O1)OC[C@@H]2[C@H]([C@@H]([C@H](C(O2)O)NC(=O)C)O)O[C@H]3[C@@H]([C@H]([C@@H]([C@H](O3)CO)O[C@H]4[C@H]([C@H]([C@@H]([C@H](O4)CO[C@@H]5[C@H]([C@H]([C@@H]([C@H](O5)CO)O)O)O[C@H]6[C@@H]([C@H]([C@@H]([C@H](O6)CO)O)O)NC(=O)C)O[C@H]7[C@@H]([C@H]([C@@H]([C@H](O7)CO)O)O)NC(=O)C)O[C@@H]8[C@H]([C@H]([C@@H]([C@H](O8)CO)O)O)O[C@H]9[C@@H]([C@H]([C@@H]([C@H](O9)CO)O)O)NC(=O)C)O)O)NC(=O)C)O)O)O

# J CASE STUDY OF MOLECULE CAPTIONING

Figure 11 shows more molecule caption examples selected from ChEBI-20 test dataset.

| Molecule Input | Atomas Output | Ground Truth |
|---|---|---|
| BLEU-2: 1.0 | The molecule is an unsaturated fatty acyl-CoA that results from the formal condensation of the thiol group of coenzyme A with the carboxy group of 8,9-epoxy-(5Z,11Z,14Z)-icosatrienoic acid. It is a long-chain fatty acyl-CoA and an unsaturated fatty acyl-CoA. It derives from an 8,9-EET. It is a conjugate acid of an 8,9-epoxy-(5Z,11Z,14Z)-icosatrienoyl-CoA(4-). | The molecule is an unsaturated fatty acyl-CoA that results from the formal condensation of the thiol group of coenzyme A with the carboxy group of 8,9-epoxy-(5Z,11Z,14Z)-icosatrienoic acid. It is a long-chain fatty acyl-CoA and an unsaturated fatty acyl-CoA. It derives from an 8,9-EET. It is a conjugate acid of an 8,9-epoxy-(5Z,11Z,14Z)-icosatrienoyl-CoA(4-). |
| BLEU-2: 0.95 | The molecule is an organophosphate oxoanion obtained by deprotonation of the carboxy and phosphate OH groups of (9S,10R)-10-hydroxy-9-(phosphonooxy)octadecanoic acid; major species at pH 7.3. It is an organophosphate oxoanion and a monocarboxylic acid anion. It is a conjugate base of a (9S,10R)-10-hydroxy-9-(phosphonooxy)octadecanoic acid. | The molecule is an organophosphate oxoanion obtained by deprotonation of the carboxy and phosphate OH groups of (9S,10R)-10-hydroxy-9-(phosphonatooxy)octadecanoic acid; major species at pH 7.3. It is an organophosphate oxoanion and a hydroxy monocarboxylic acid anion. It is a conjugate base of a (9S,10R)-10-hydroxy-9-(phosphonooxy)octadecanoic acid. |
| BLEU-2: 0.9 | The molecule is a methyl glycoside that consists of a 4-O-(5-aminopentyl)-alpha-D-mannose residue and three N-formyl-alpha-D-perosamine residues linked sequentially (1->2), (1->3) and (1->2) and linked at the reducing end glycosidically to a methyl group. It is a methyl glycoside and a trisaccharide derivative. | The molecule is a methyl glycoside that consists of a 4-O-(5-aminopentyl)-alpha-D-mannose residue and two N-formyl-alpha-D-perosamine residues linked sequentially (1->2) and (1->3) and linked at the reducing end glycosidically to a methyl group. It is a methyl glycoside and a trisaccharide derivative. |
| BLEU-2: 0.85 | The molecule is the organosulfonate oxoanion that is the trianion of Reactive Blue 5, formed by loss of a proton from each of the sulfo groups; major species at pH 7.3. It is a conjugate base of a Reactive Blue 5. | The molecule is the organosulfonate oxoanion that is the trianion of Reactive Blue 5 quinol form, obtained by loss of a proton from each of the sulfo groups; major species at pH 7.3. It is a conjugate base of a Reactive Blue 5 quinol form. |
| BLEU-2: 0.8 | The molecule is a pyrrolizine alkaloid that is produced by a hybrid species of Jacobaea. It has a role as a Jacobaea metabolite. It is a pyrrolizine alkaloid, a tertiary amine oxide, a tertiary alcohol, a macrocyclic lactone, an organic heterotricyclic compound and a tertiary amine oxide. It derives from a Senecivernine. | The molecule is a pyrrolizine alkaloid that is jacoline in which the tertiary amino function has been oxidised to the corresponding N-oxide. It has a role as a Jacobaea metabolite. It is a macrocyclic lactone, an organic heterotricyclic compound, a pyrrolizine alkaloid, a triol and a tertiary amine oxide. It derives from a jacoline. |
| BLEU-2: 0.7 | The molecule is a pyrrolizine alkaloid that is produced by a hybrid species of Jacobaea. It has a role as a Jacobaea metabolite. It is a pyrrolizine alkaloid, a tertiary amine oxide, a tertiary alcohol, a macrocyclic lactone, an organic heterotricyclic compound and a tertiary amine oxide. It derives from a Senecivernine. | The molecule is a pyrrolizine alkaloid that is jacoline in which the tertiary amino function has been oxidised to the corresponding N-oxide. It has a role as a Jacobaea metabolite. It is a macrocyclic lactone, an organic heterotricyclic compound, a pyrrolizine alkaloid, a triol and a tertiary amine oxide. It derives from a jacoline. |
| BLEU-2: 0.6 | The molecule is a hydroxy fatty acid anion obtained by deprotonation of the carboxy function of lipoxin A4; major species at pH 7.3. It has a role as a human metabolite and a Saccharomyces cerevisiae metabolite. It is a hydroxy fatty acid anion, an icosanoid anion, a long-chain fatty acid anion and a polyunsaturated fatty acid anion. It is a conjugate base of a lipoxin A4. | The molecule is a hydroxy fatty acid anion obtained by deprotonation of the carboxy function of lipoxin A4: major species at pH 7.3. It is a hydroxy fatty acid anion and a lipoxin anion. It is a conjugate base of a lipoxin A4. |

Figure 11: **Additional molecule captioning examples.**

# K    CASE STUDY OF TEXT-BASED DE NOVO MOLECULE GENERATION

Figure 12 shows more text-based de novo molecule generation examples selected from ChEBI-20 test dataset.

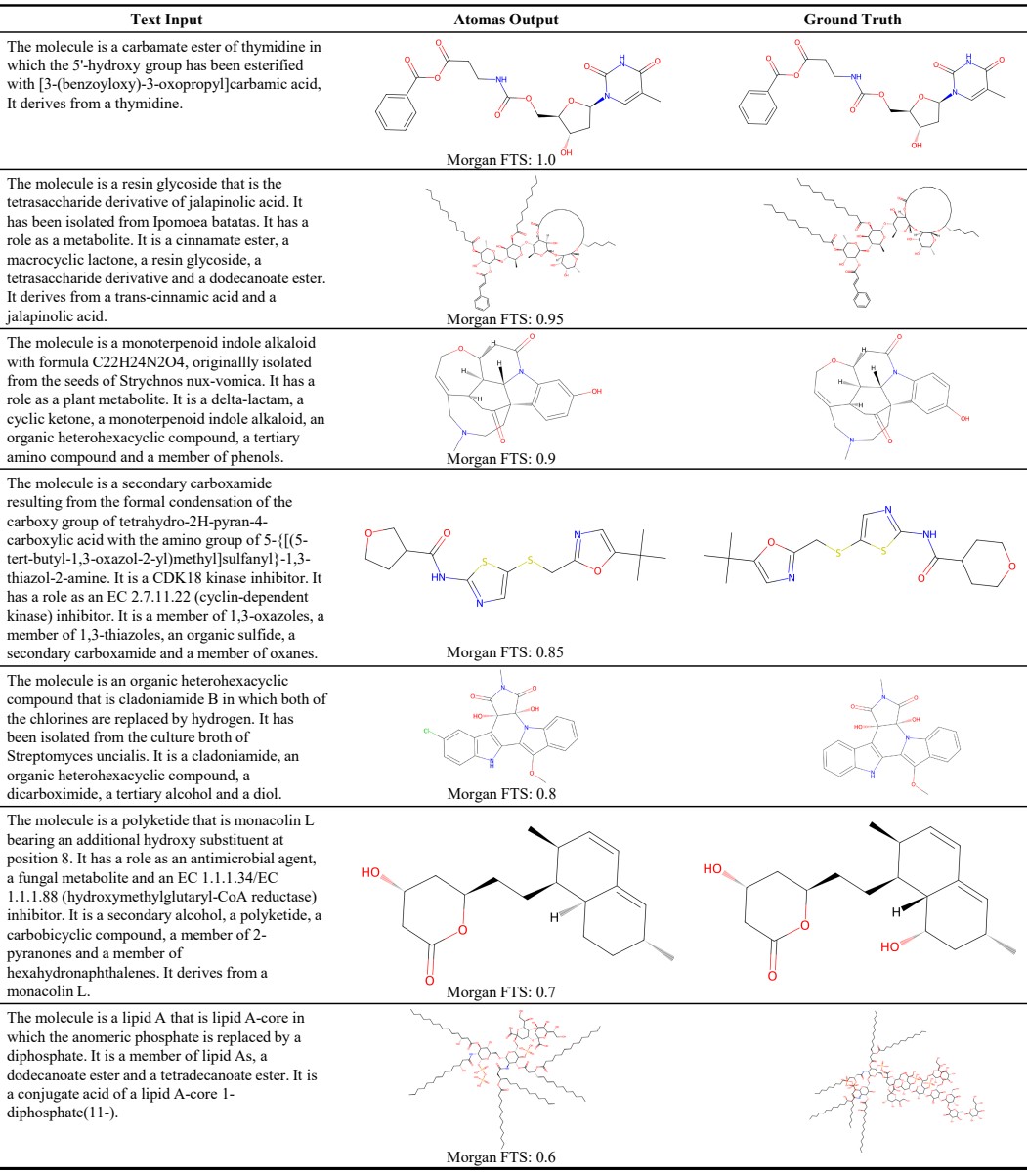

Figure 12: **Additional text-based de novo molecule generation examples.**

