# OpenReview forum: "Atomas: Hierarchical Adaptive Alignment on Molecule-Text for Unified Molecule Understanding and Generation"
_ICLR.cc/2025/Conference — ICLR 2025 Poster_

### Official Review · Reviewer_1BNJ · 2024-10-22

**Soundness:** 3
**Presentation:** 3
**Contribution:** 3
**Rating:** 6
**Confidence:** 5

**Summary:**

This paper presents a hierarchical molecular representation learning framework, namely Atomas, which jointly learns representations from SMILES strings and texts. In Atomas, a Hierarchical Adaptive Alignment model is designed to learn the fine-grained fragment correspondence between molecule SMILES strings and text descriptions at three semantic levels, namely atom, fragment, and molecule level. Atomas outperform the baseline models on 12 molecule-related tasks.

**Strengths:**

1. The idea of aligning molecules and texts in a fine-grained aspect is novel and provides insight into the methodology for adopting LLMs for molecule discovery.
2. The module, Hierarchical Adaptive Alignment, is novel and introduces a new modality alignment method.
3. The performance of Atomas is competitive and proves its generalization in different molecule-related tasks.

**Weaknesses:**

1. The description of the framework is inappropriate. The authors claim they align SMILES and textual descriptions in three semantic levels, but the atom level actually uses tokens, which can not be identical to `atoms` as symbols and numbers might exist in the SMILES strings.
2. In Table 3, the performance of MolCA is not correct. The metrics should be BLEU-2 63.9, BLEU-4 55.5, ROUGE-1 69.7, ROUGE-2 55.8, ROUGE-L 63.6, and METEOR 66.9 with Galac1.3B [1]. This raises a challenge to the authors' claim that these molecule-and-text alignment methods struggle to effectively capture fine-grained correspondence related to local parts within different modalities.
3. I am a little confused about the three-level alignment. It seems that the alignment still happens at Stage 3, which calculates the similarity between the textual description and the molecule representation. Although the token and token cluster information might be extracted to enhance the molecule representations, I do not see how atom or fragment information is aligned with their descriptions in a fine-grained manner.
4. The selected baseline models are probably weak. More advanced models like BioT5 [2] [3] and ICMA [4] should be included.


#### References
[1] Liu, Z., Li, S., Luo, Y., Fei, H., Cao, Y., Kawaguchi, K., ... & Chua, T. S. (2023). MolCA: Molecular graph-language modeling with cross-modal projector and uni-modal adapter. arXiv preprint arXiv:2310.12798.
[2] Pei, Q., Zhang, W., Zhu, J., Wu, K., Gao, K., Wu, L., ... & Yan, R. (2023). Biot5: Enriching cross-modal integration in biology with chemical knowledge and natural language associations. arXiv preprint arXiv:2310.07276.
[3] Pei, Q., Wu, L., Gao, K., Liang, X., Fang, Y., Zhu, J., ... & Yan, R. (2024). Biot5+: Towards generalized biological understanding with iupac integration and multi-task tuning. arXiv preprint arXiv:2402.17810.
[4] Li, J., Liu, W., Ding, Z., Fan, W., Li, Y., & Li, Q. (2024). Large Language Models are In-Context Molecule Learners. arXiv preprint arXiv:2403.04197.

**Questions:**

1. Could the authors compare more advanced baselines?
2. Could the authors explain my concerns in Weakness 3?
3. Is it possible to scale this methodology further or apply Atomas to decoder-only models? Currently, it might be unfair to compare Atomas with the previous baselines like MolCA because the backbone LLM can also lead to a difference in performance.

---

> ### Author Response · Authors · 2024-11-20
>
> We thank reviewer 1BNJ for the valuable feedback and for recognizing the innovation and effectiveness of our method. Please kindly find the detailed responses point-by-point below. Any further comments and discussions are welcomed!
>
>
> **W1: The description of atom level.**
>
> **Reply:** Thank you for your insightful feedback. We appreciate your concern regarding the terminology used to describe the "atom level" alignment. Our intention with the term "atom level" was to refer to alignment at the smallest unit of division within the molecule. However, we acknowledge that the inclusion of numbers in SMILES strings, which are not strictly atoms, may lead to potential confusion.
>
> To address this, we will revise the terminology in our paper to "word level," which we believe more accurately reflects the nature of the alignment process while avoiding ambiguity.
>
> **W2: The report metrics of MolCA.**
>
> **Reply:** Thank you for your careful review and for bringing up the performance metrics of [1] in `Table 3`. We would like to clarify that **the metrics you referred to are from an earlier version of [1]**. Our reported metrics are based on the updated versions of [1], as reflected in their arXiv versions V3 or V4 and ACL Anthology version V2. If it is convenient, we encourage you to check these updated sources for the latest results. We have included the latest MolCA paper below for your reference.
>
> With this clarification, we would like to emphasize that our method achieves superior performance while relying solely on SMILES, in contrast to models that depend on both SMILES and graph representations for certain tasks.
>
> [1] MolCA: Molecular Graph-Language Modeling with Cross-Modal Projector and Uni-Modal Adapter. EMNLP, 2023.
>
> **W3&Q2: Confusion about the alignment process.**
>
> **Reply:** Thank you for your thoughtful feedback. We would like to clarify how the three-level alignment works in our Hierarchical Adaptive Alignment (HAA) model. As illustrated in `Figure 2` and the `Algorithm` provided in `Appendix D` of our paper, the alignment between molecules and text is performed at three levels: atom, fragment, and molecule.
>
> At the **fragment level**, the process consists of three steps:
>
> 1. **Step1: Assignment Step**: Tokens at the atom level are assigned to form clusters.
> 2. **Step2: Merge Step:** These clusters are further merged to generate the fragment representations.
> 3. **Step3: Alignment Step:** The Weighted Alignment Module (WAM) aligns the fragments between the two modalities.
>
> At the molecule level, fragments are further clustered into higher-level representations using the same Assignment and Merge steps. The WAM then aligns these higher-level molecule representations between the text and molecule modalities.
>
> In summary:
>
> * **Steps 1 and 2 (Assignment and Merge)** are executed twice—once at the fragment level and once at the molecule level.
> * **Step 3 (Alignment)** is executed three times—once at each level: atom, fragment, and molecule.
>
> By iteratively applying steps 1 and 2 along with the WAM, we construct a hierarchical alignment structure that spans three levels of abstraction, enabling the model to capture fine-grained alignments across varying levels of detail.
>
> We hope this explanation clarifies how our three-level alignment works and addresses your concerns.

---

> ### Author Response · Authors · 2024-11-20
>
> **W4&Q1: The compared baseline models.**
>
> **Reply:** Thank you for suggesting. We believe the baseline models we have included represent the latest published methods and demonstrate strong performance. However, we agree that ICMA is a relevant baseline and have incorporated a comparison with ICMA, as shown in the table below. We will also include a citation to ICMA in the revised version of the paper.
>
>
> |           Model            | Params |     BLEU↑      | Levenshtein↓  |   MACCS FTS↑   |    RDK FTS↑    |  Morgan FTS ↑  | Validity↑  |
> |:--------------------------:|:------:|:--------------:|:-------------:|:--------------:|:--------------:|:--------------:|:----------:|
> | ICMA(Galactica-125M)2,2048 |  125M  |     0.836      |     21.48     |     0.893      |     0.809      |     0.743      |   0.825    |
> |   ICMA(Mistral-7B)4,2048   |   7B   |     0.855      |     18.73     |     0.916      |     0.837      |     0.789      |   0.891    |
> |        Atomas-base         |  271M  |     0.868      |     13.76     |     0.908      |     0.827      |     0.773      |   0.971    |
> |        Atomas-large        |  825M  | **0.874±.003** | **12.70±.28** | **0.914±.004** | **0.841±.002** | **0.788±.002** | **0.980±.003** |
>
> Based on the data presented, although Atomas-large has only approximately 12% of the parameters of ICMA, it still achieves superior performance.
>
> Regarding BioT5 and BioT5+, we acknowledge their use of SELFIES as a molecular representation, which ensures 100% validity and contributes to their reported exactness (0.413) and validity (1.00) metrics. However, we attribute these advantages to the choice of molecular representation rather than methodological differences. Since SMILES is the dominant molecular representation, Atomas focuses on SMILES and compares only with methods that also use SMILES. Despite the exactness and validity metrics of SELFIES-based methods, Atomas achieves superior performance in quantitative metrics compared to the BioT5 and BioT5+, as shown in the table below:
>
>
> |    Model     |     BLEU↑      | Levenshtein↓  |   MACCS FTS↑   |    RDK FTS↑    |  Morgan FTS ↑  |
> |:------------:|:--------------:|:-------------:|:--------------:|:--------------:|:--------------:|
> |    BioT5     |     0.867      |    15.097     |     0.886      |     0.801      |     0.734      |
> |    BioT5+    |     0.872      |    12.776     |     0.907      |     0.835      |     0.779      |
> | Atomas-large | **0.874±.003** | **12.70±.28** | **0.914±.004** | **0.841±.002** | **0.788±.002** |
>
> Additionally, we have already included qualitative comparisons with BioT5 in Figures 6 and 7, which analyze molecule captioning and generation tasks. These figures show that global alignment methods like BioT5 struggle to differentiate between enantiomers such as "D-glutamate" and "L-glutamate." In contrast, Atomas generates more accurate and detailed molecular descriptions, highlighting the effectiveness of our hierarchical alignment model.
>
> **Q3: Scaling Atomas to the decoder-only models.**
>
> **Reply:** Thank you for your insightful feedback. We believe that the Atomas Hierarchical Adaptive Alignment can indeed be applied to decoder-only models as a plug-in module. The hierarchical alignment can be performed directly on the representations of the decoder, further highlighting the versatility and adaptability of our approach. In alignment with the goal expressed in our title, our aim was to develop a unified model capable of addressing both understanding tasks (via the encoder) and generative tasks (via the encoder-decoder combination).
>
> Regarding the choice of backbone LLM, we respectfully believe that our comparison remains fair, as each method employs distinct strategies for molecule-text alignment, input modalities, and model architectures. Moreover, the parameter sizes of the models vary substantially. For example, ICMA employs a 7B parameter backbone, while Atomas utilizes fewer than 1B parameters. Given these differences, ensuring identical backbone LLMs across all methods in the comparison may not always be feasible or practical. To provide additional clarity, we have discussed the variations among existing molecule-text alignment methods in Appendix A.1 (Table 10), detailing their input formats, alignment strategies, training approaches, and application tasks.
>
>
> To ensure complete fairness, we have used the same backbone LLM as MolT5 in our experiments. As shown in Table 3 of the paper, our Atomas-base model outperforms the MolT5-large model while using only 35% of its parameters and without initial pretraining. This highlights the efficiency and effectiveness of our proposed framework. We will further emphasize this point in the revised version of the paper.

---

> ### Comment · Reviewer_1BNJ · 2024-11-20
>
> I want to first thank the authors for their responses. The rebuttal has already adressed some of my concerns.
>
>  However, the performance on the molecule-caption translation task is only on the ChEBI-20 dataset, which can not reveal the generalization of the proposed method regarding this downstream task. Meanwhile, the performance of Atomas is still close to the previous baselines. I would like to recommend that the authors add the Mol-Instruction [1] and PubChem [2] datasets for further demonstration. I am happy to increase my score if we see similar improvements on these datasets.
>
> #### **References**
> [1] Fang, Y., Liang, X., Zhang, N., Liu, K., Huang, R., Chen, Z., ... & Chen, H. (2023). Mol-instructions: A large-scale biomolecular instruction dataset for large language models. arXiv preprint arXiv:2306.08018.
> [2] Liu, Z., Li, S., Luo, Y., Fei, H., Cao, Y., Kawaguchi, K., ... & Chua, T. S. (2023). MolCA: Molecular graph-language modeling with cross-modal projector and uni-modal adapter. arXiv preprint arXiv:2310.12798.

---

> ### Author Response · Authors · 2024-11-24
> **Addtional experiments on molecule-caption translation task**
>
> We are encouraged by `Reviewer 1BNJ`'s positive feedback and agree that adding additional experiments can better showcase Atomas' generalization ability.
>
> In response to your suggestion, we conducted an additional experiment to compare Atomas' performance with the state-of-the-art model, ICMA, on the PubChem dataset provided by MolCA. Specifically, we trained Atomas-large using only the PubChem324K training dataset, aligning the experimental settings with ICMA by ignoring pretaining data. The results are presented in the table below.
>
> *Supplementary experiment for Cap2Mol results on PubChem324k test dataset*
> |           Model            | Params |   BLEU↑   | Levenshtein↓ | MACCS FTS↑ | RDK FTS↑  | Morgan FTS ↑ | Validity↑ |
> |:--------------------------:|:------:|:---------:|:------------:|:----------:|:---------:|:------------:|:---------:|
> | ICMA(Galactica-125M)2,2048 |  125M  |   0.569   |    52.75     |   0.719    |   0.579   | 0.652   |   0.825   |
> |   ICMA(Mistral-7B)4,2048   |   7B   |   0.450   |    77.01     |   0.764    |   0.624   |    0.504     |   0.891   |
> |        Atomas-large        |  825M  | 0.734 |  28.186  | 0.773  | 0.637 | 0.535  | 0.945 |
>
> Based on the data presented, Atomas achieves significant improvements on most metrics over ICMA, including a 63.11% improvement on the BLEU metric and a 46.54% improvement on Levenshtein compared to ICMA. This demonstrates Atomas' superior performance on the PubChem dataset, highlighting its robust generalization capabilities. Given the extensive size of the Mol-Instruction dataset (297K training samples vs. 12K training samples in PubChem) challenge for us during the rebuttal period, we prioritized the PubChem dataset during the rebuttal to demonstrate Atomas's generalization within the available timeframe.
>
> We hope this response addresses your concerns. Please let us know if you have any further questions or concerns.

---

> > ### Comment · Reviewer_1BNJ · 2024-11-24
> >
> > Thanks for your responses. I will raise my score.

---

> > > ### Author Response · Authors · 2024-11-24
> > > **Thanks for your acknowledgment and improvement of rating**
> > >
> > > Dear `Reviewer 1BNJ`,
> > >
> > > We appreciate your recognition of our efforts and are very glad you increased the rating!
> > >
> > > Thanks again for your valuable suggestions and comments.

---

### Official Review · Reviewer_N9hg · 2024-11-03

**Soundness:** 3
**Presentation:** 3
**Contribution:** 2
**Rating:** 6
**Confidence:** 5

**Summary:**

This paper proposed a unified framework to realize the alignment between text and SMILES at three levels without fine-grained manually annotated labels, namely the atom level, fragment level, and molecule level. The combined optimization of fine-grained contrastive learning and autoregressive learning promotes the model's performance on various downstream tasks, such as text-based de novo molecule generation, molecule captioning, molecule property prediction and molecule-text retrieval.

**Strengths:**

1.The writing is clear, the diagrams are rich, the experimental results are comprehensive, and it is easy to understand.
2. This paper realizes the fine-grained alignment of SMILES and text from the atom level, fragment level and molecule level, which is very comprehensive. The visualized results demonstrate the effectiveness of fine-grained alignment.
3. The ablation experiments demonstrate a mutual enhancement effect: contrastive learning alignment improves autoregressive learning, while autoregressive learning enhances contrastive learning alignment, showing the benefit of joint optimization.
4. Atomas, relying only on SMILES, achieves superior performance than models that rely on both SMILES and graph on some tasks.

**Weaknesses:**

1.Previous papers have proved the effectiveness of combined optimization of contrastive learning alignment and autoregressive learning.
Ref.”Align before fuse: Vision and language representation learning with momentum distillation”
2. The practice of using all tokens to calculate similarity to achieve more fine-grained alignment without additional labels is also not new. And the direct application of the weighted alignment modules limits the innovation.
Ref.”FILIP: Fine-grained Interactive Language-Image Pre-Training”、 “Disentangled Representation Learning for Text-Video Retrieval”.
3. While the method of constructing hierarchical features in this paper is effective, it lacks a more explicit rationale explaining the underlying principles that make this construction feasible.

**Questions:**

1. It seems that the purpose of the Molecule Level Alignment in Hierarchical Adaptive Alignment, and the Global Alignment is the same. Can you explain more about why Global Alignment is still needed in the architecture?
2. In the Assignment Step, have you tried using other clustering algorithms? How will it affect the results?

---

> ### Author Response · Authors · 2024-11-20
>
> We sincerely thank Reviewer N9hg for their valuable feedback and for recognizing the comprehensive methodology and experiments, the intuitive and clear presentation, and the extensive ablation studies contributed by our method. Please find our detailed point-by-point responses below. We welcome any further comments and discussions!
>
> **W1: Contrastive-autoregressive synergy boosts representation has been proved.**
>
> **Reply:** We sincerely thank the reviewer for their insightful comment and for referencing [1]. We have cited [1] in our paper, as it inspired us to adopt the use of soft labels. However, we respectfully note that [1] primarily demonstrates the effectiveness of the combined optimization of `contrastive learning alignment` and `autoencoder learning`, rather than `autoregressive learning`. Specifically, the pre-training objectives in [1] are contrastive learning and masked language modeling, distinguishing its focus from the contrastive-autoregressive optimization explored in our work.
>
> To further clarify the effectiveness of the contrastive-autoregressive synergy in boosting representation, we provide the following in-depth discussion of this joint optimization strategy:
>
> * **Model Perspective:** As suggested by the existing study [2], attention-based generation tasks essentially perform a form of soft alignment. During the generation process, the attention mechanism facilitates a mutual translation between text and SMILES, reinforcing the semantic consistency between the textual description and the molecular structure it represents. Concurrently, representation learning bridges the domain gap between text and SMILES, enhancing the caption/generation task.
>
> * **Data Perspective:** The captioning and generation tasks may provide complementary information for learning molecular representations. These tasks necessitate the model to learn the mapping between text and molecular domains, which allows the model to grasp the intricate relationship between textual descriptions and molecular structures, thereby enriching the quality of the learned molecular representations. Hierarchical alignment further aids in capturing local data pair relationships, benefiting the generation process.
>
> In summary, we believe that Atomas provides new insights into **molecule generation tasks** through its joint optimization strategy. This approach enables the model to learn more effective molecular representations by leveraging the complementary information provided by each task. The experimental results presented in Table 8 and Figure 4 of the paper further substantiate the effectiveness of this strategy.
>
> [1] Align before fuse: Vision and language representation learning with momentum distillation. NeurIPS 2021, Junnan Li.
>
> [2] Neural Machine Translation by Jointly Learning to Align and Translate. ICLR 2015, oral, Yoshua Bengio.
>
> **W2: Using all tokens to calculate similarity and the use of the weighted alignment module.**
>
> **Reply:** Thank you for your thoughtful feedback. In our method, similarity is calculated using all tokens only at the first level: the atom level. As acknowledged in our paper, we employ the Weighted Alignment Module (WAM) inspired by [1] to perform alignment.
>
> However, we would like to highlight an important distinction. The weighted alignment in [1, 2] primarily focus on alignment at a single level, utilizing `token-wise interactions`. In contrast, our approach extends beyond token-wise interactions by introducing a hierarchical structure. This enables us to perform alignment in a `set-wise manner`, where groups of semantically similar tokens interact. Furthermore, to the best of our knowledge, we are the first to apply such an alignment manner specifically in the text-to-molecule domain. We are also actively exploring alternative alignment modules tailored for set-wise interactions to further enhance the novelty and effectiveness of our framework. Thank you again for your insightful comments!
>
> [1] Disentangled Representation Learning for Text-Video Retrieval. arxiv 2022.
>
> [2] FILIP: Fine-grained Interactive Language-Image Pre-Training. ICLR 2022.

---

> ### Author Response · Authors · 2024-11-20
>
> **W3: The hierarchical feature construction is effective but needs more explicit rationale explaining.**
>
> **Reply:** Our approach is motivated by the observation that text-molecule multimodal data naturally exhibits a hierarchical structure. For instance, textual descriptions and molecular representations can both be understood at multiple levels of granularity, such as **word-atom, phrase-fragment, and paragraph-molecule**. Recognizing this inherent hierarchy inspired us to adopt hierarchical clustering [1] for fine-grained alignment. we employ a "bottom-up" strategy, where each observation begins in its own cluster, and clusters are progressively merged as the hierarchy ascends.
>
> To further validate the rationale for our approach, we have provided a visual example in Figure 5 of the paper. Figure 5 illustrates how our method constructs hierarchical features. The molecule shown in Figure 5 is formed by combining atoms at positions 0–15 and 16–26 through dehydration condensation. At the fragment level, our model clusters atoms (or words) into functional groups (or phrases) using the Adaptive Polymerization Module. As the hierarchy progresses to the molecule level, fragments are further clustered to form monomer-like structures.
>
> Additionally, the performance improvements demonstrated in the ablation study in Table 7 further validate the effectiveness and practicality of this hierarchical feature construction. These empirical results underscore the value of modeling the hierarchical structure inherent in multimodal molecular data.
>
> [1] Algorithms for hierarchical clustering: an overview. Wiley Interdisciplinary Reviews: Data Mining and Knowledge Discovery 2011.
>
> **Q1: The reason for using both global alignment and molecule level alignment.**
>
> **Reply:** Thank you for your thoughtful feedback. We would like to clarify that these two alignments serve distinct purposes in our architecture. The Global Alignment and Hierarchical Adaptive Alignment ensures that the model captures both local features and global information effectively.
>
> In Global Alignment, we aggregate all tokens to obtain the global feature $(bsz, 1)$, where $bsz$ represents the batch size. This is akin to the use of the $[CLS]$ token in BERT-like models to capture global information.
>
> In contrast, Molecule Level Alignment operates at a higher level of granularity. As illustrated in Figure 2 of the paper, molecular fragments, and textual paragraphs are clustered into higher-level groups. At this level, tokens are aggregated into $(bsz, n_{s})$ and $(bsz, n_{t})$, where $n_{s}$ and $n_{t}$ denote the number of molecular and text clusters, respectively. For example, as shown in Figure 2, alignment is performed between 2 molecular clusters and 1 text cluster.
>
> We name this the "Molecule Level" alignment because, at this stage, the model is capable of capturing relationships between complex molecules composed of multiple submolecular units. For example, as depicted in Figure 5, Atomas effectively clusters the *trans-sinapic acid* and *beta-D-glucose* into distinct classes, reflecting their independent molecular identities despite being part of a larger structure. This ability to recognize and align such complex molecular components is a unique feature of our approach.
>
> **Q2: Try using other clustering algorithms.**
>
> **Reply:** Thank you for your thoughtful question. Yes, we have explored general clustering algorithms, including general approaches such as k-means clustering and molecular-specific methods like BRICS decomposition.
> 1. **BRICS Decomposition:**
> We applied the BRICS decomposition method to fragment the molecules shown in Figure 5. The fragments produced by BRICS include:
> * [1*]C(=O)C[7*]
> * [13*][C@@H]1OC@HC@@HC@H[C@H]1O
> * [16*]c1cc([16*])c(O)c([16*])c1
> * [3*]OC
> * [3*]O[3*]
> * [7*]C[8*]
> * [8*]CO
>
> However, BRICS introduces characters not present in the original SMILES strings and does not allow for hierarchical decomposition, limiting its direct replacement with the Adaptive Polymerization Module in Atomas. Additionally, as discussed in [1], BRICS typically generates only a few fragments for each molecule, resulting in coarse-grained representations that may lack the level of detail required for effective language modeling.
>
> 2. **K-Means Clustering:**
> We also considered k-means clustering. However, this method requires specifying the number of clusters in advance. Unlike images with fixed dimensions (e.g.,$512*512$), molecules and text exhibit variable lengths in their SMILES and textual representations, leading to varying cluster requirements. Therefore, predefining a fixed number of clusters is not practical for molecular data. Instead, we chose to use the density peak-based clustering algorithm, which dynamically assigns tokens to clusters without requiring a predefined cluster count.
>
> [1] Multi-objective drug design based on graph-fragment molecular representation and deep evolutionary learning. Frontiers in Pharmacology 2022.

---

> > ### Author Response · Authors · 2024-11-24
> > **Would you mind checking our responses and confirming if you have any further questions?**
> >
> > Dear `Reviewer N9hg`,
> >
> > Thanks very much for your time and valuable comments.
> >
> > In the rebuttal period, we provided detailed responses to all your comments and questions point-by-point for the unclear presentations.
> >
> > Would you mind reviewing our responses and letting us know if you have any additional questions or concerns?
> >
> > We hope our rebuttal has effectively addressed your concerns, and we look forward to your feedback!

---

> > > ### Comment · Reviewer_N9hg · 2024-11-26
> > >
> > > Thanks for your response which clarifies some of my concerns. I will keep my score.

---

> > > > ### Author Response · Authors · 2024-11-27
> > > > **Thank you for your response**
> > > >
> > > > Thank you for your feedback and for taking the time to review our rebuttal. We appreciate your valuable comments and insights.

---

### Official Review · Reviewer_1nxB · 2024-11-03

**Soundness:** 3
**Presentation:** 3
**Contribution:** 3
**Rating:** 6
**Confidence:** 4

**Summary:**

This paper notices that existing works about molecule-and-text representation learning are mainly coarse-grained and uses a global alignment method to integrate molecules and texts. To solve this problem, this paper proposes a fine-grained method for cross-modal representation learning. Specifically, this paper proposes a hierarchical adaptive alignment module, which consists of two components, adaptive polymerization module and weighted alignment module. Experiments on different cross-modal tasks verify the effectiveness of the proposed model. Ablation analysis is also conducted to show the effect of each modeling component.

**Strengths:**

1. Overall, the problem of existing works is clearly motivated in the Introduction section with a figure as visual illustration. The overall writing in the paper is also clear enough, and an algorithm is also provided to formally present the learning process.

2. Specifically, the design of hierarchical adaptive alignment module is interesting and novel. This paper clearly shows its effect with empirical evaluation.

3. Experiments are comprehensive enough with different tasks and metrics. Ablation analysis is also conducted. Visualization also helps understand what the model learns.

**Weaknesses:**

1. Usually when we do experiments, we encourage authors to repeat the same experimental setting multiple times and report both mean and standard deviation, or report significance t-test. However, some tables and figures in the Experiment section don't have standard deviation or significance t-test, such as Tables 1 and 8, Figure 4. Authors are suggested provide standard deviation in the paper.

2. Scalability is an experiment in the paper to show the proposed is possible to scale on large datasets. For scalability, authors are also suggested to provide computational comeplexity to theoretically show that the proposed model is computationally efficient than baseline models.

**Questions:**

N.A.

---

> ### Author Response · Authors · 2024-11-20
>
> We thank reviewer 1nxB for the valuable and positive feedback and for acknowledging the novelty, clear present, and adequate ablation contributed by our method. We have addressed all the comments. Please kindly find the detailed responses below. Any further comments and discussions are welcomed!
>
> **W1: Missing the standard deviation or significance t-test in Retrieval Tasks.**
>
> **Reply:** Thanks for your valuable comment. In the `retrieval task`, as all models are evaluated **without fine-tuning**, the varying seeds do not influence the experimental results, leading to the standard deviation equal to 0.
>
> We greatly appreciate your suggestion and acknowledge the importance of including such details to improve the rigor of our work. To address your concern, we have re-pretrained Atomas-base and Atomas-large three times and report the standard deviation results in below tables:
>
> *Supplementary experiment for Text to Molecule Task in Table 1:*
> | Method (No Fine-tuning) |          R@1           |        R@5         |        R@10        |         MRR          |
> |:-----------------------:|:----------------------:|:------------------:|:------------------:|:--------------------:|
> |     **Atomas-base**     |  39.07（$\pm$ 0.03）   | 59.70 ($\pm$ 0.06) | 66.57 ($\pm$ 0.11) |  47.32($\pm$ 0.02)   |
> |    **Atomas-large**     | 49.08  （$\pm$ 0.02） | 68.35($\pm$ 0.05)  | 73.13 ($\pm$ 0.07) | 57.80   ($\pm$ 0.02) |
>
>
> *Supplementary experiment for Molecule to Text Task in Table 1:*
> | Method (No Fine-tuning) |         R@1         |        R@5         |        R@10        |         MRR          |
> |:-----------------------:|:-------------------:|:------------------:|:------------------:|:--------------------:|
> |     **Atomas-base**     | 37.86（$\pm$ 0.05） | 59.24 ($\pm$ 0.04) | 65.56 ($\pm$ 0.04) |  47.80($\pm$ 0.01)   |
> |    **Atomas-large**     | 46.20（$\pm$ 0.03） | 66.05($\pm$ 0.03)  | 72.30 ($\pm$ 0.06) | 55.52   ($\pm$ 0.02) |
>
>
> Regarding Table 8 and Figure 4, we would like to clarify that these are **ablation studies** specifically designed to highlight the influence of model setting choices rather than variability in task performance. Nonetheless, we have included the standard deviation results for Table 8 in the tables below:
>
> *Supplementary experiment for Text to Molecule Task in Table 8:*
> |   Training Strategy    |         R@1         |        R@5         |        R@10        |        MRR        |
> |:----------------------:|:-------------------:|:------------------:|:------------------:|:-----------------:|
> |        2Stages         | 37.71（$\pm$ 0.02） | 58.00 ($\pm$ 0.02) | 65.02 ($\pm$ 0.09) | 45.91($\pm$ 0.04) |
> | **Joint optimization** |  39.07（$\pm$ 0.03  | 59.70 ($\pm$ 0.06) | 66.57 ($\pm$ 0.11) | 47.32($\pm$ 0.02) |
>
> *Supplementary experiment for Molecule to Text Task in Table 8:*
> |   Training Strategy    |         R@1         |        R@5         |        R@10        |        MRR        |
> |:----------------------:|:-------------------:|:------------------:|:------------------:|:-----------------:|
> |        2Stages         | 36.52（$\pm$ 0.03） | 57.30 ($\pm$ 0.01) | 63.52 ($\pm$ 0.07) | 46.07($\pm$ 0.03) |
> | **Joint optimization** |  37.86（$\pm$ 0.05)  | 59.24 ($\pm$ 0.04) | 65.56 ($\pm$ 0.04) | 47.80($\pm$ 0.01) |
>
> We will also include these standard deviation results in our revised version of the paper.

---

> ### Author Response · Authors · 2024-11-20
>
> **W2: Adding theoretically computational comeplexity.**
>
> **Reply:** Thank you for your valuable suggestion. The time complexity for the global alignment is $\mathcal{O}(D)$, while for fine-grained alignment it is $\mathcal{O}(N_{t}*N_{s}*D)$, where $N_{t}$ and $N_{s}$ represent the number of text tokens and SMILES tokens, respectively, and $D$ is the dimensionality of the representations. We will incorporate this computational complexity analysis into the revised version of the paper.
>
>
> To provide more clarity, the following table (also presented as Table 7 in the paper) compares global alignment (GA) and hierarchical alignment (HA) in Atomas-base. We report the average training time on the Chebi-20 dataset using an NVIDIA A100 GPU, along with performance metrics on the molecule generation task.
>
> |         **Method**          | **Training Time(s/sample)** | **Exact↑** | **Levenshtein↓** | **Morgan FTS↑** |
> |:---------------------------:|:------------:|:----------:|:----------------:|:---------------:|
> |    Global Alignment (GA)    |    0.0119    |   0.223    |      16.946      |      0.716      |
> | Hierarchical Alignment (HA) |    0.0132    |   0.266    |      16.675      |      0.736      |
> |            GA+HA            |    0.0145    |   0.298    |      15.472      |      0.75       |
>
> We acknowledge that fine-grained alignment introduces additional computational overhead. However, we believe the observed performance improvements shown in the table above justify this marginal increase in computational cost.
>
> Additionally, we have made every effort to include statistics on the pre-training computational cost for baseline methods, as shown in the table below. However, we note that most methods do not explicitly report pre-training time costs. Variations in batch sizes, GPU types, and other factors across different setups make it challenging to estimate these costs with high accuracy.
>
> *Supplementary experiment for computational time comparisons:*
>
> |  **Model**   | **# Params**  |               **Pre-train**               | **Epochs** |     **GPUs**     | Time  |
> |:------------:|:-------------:|:-----------------------------------------:|:----------:|:----------------:|:-----:|
> | Atomas-base  |     271M      |                Full train                 |    100     |   8 A100 40GB    | 10.6h |
> | Atomas-large |     825M      |                Full train                 |     50     |   8 A100 40GB    |  11h  |
> |    MolCA     |     1.3B      | Stage1(Full train)+Stage2(freeze base LM) | 60(50+10)  |   2 A100 40GB    |  27h  |
> | InstructMol  |     6.9B      |              freeze base LM               |     5      |   4 A6000 48GB   |   -   |
> |     MoMu     | 82M/252M/782M |                Full train                 |    300     |   8 V100 32GB    |   -   |
> |    MolFM     |     61.8M     |                Full train                 |    300     | 4 A100 unknownGB |   -   |
> |    MolXPT    |     350M      |                Full train                 | 200k steps | 8 A100 unknownGB |   -   |
>
> Based on the data presented, we believe the pre-training stage of our framework remains computationally efficient and affordable. We will add these supplementary experiments to the revised version of the paper.

---

> > ### Author Response · Authors · 2024-11-24
> > **Would you mind checking our responses and confirming if you have any further questions?**
> >
> > Dear `Reviewer 1nxB`,
> >
> > Thanks very much for your time and valuable comments.
> >
> > In the rebuttal period, we provided detailed responses to all your comments and questions point-by-point for the unclear presentations. Specifically, we included：
> >
> > * Standard deviation in the retrieval tasks and the ablation study in Tables 1 and 8.
> > * Theoretical computational complexity and computational time comparisons across baselines.
> >
> > Would you mind reviewing our responses and letting us know if you have any additional questions or concerns?
> >
> > We hope our rebuttal has effectively addressed your concerns, and we look forward to your feedback!

---

> > > ### Author Response · Authors · 2024-12-02
> > > **Looking Forward to Your Valuable Feedback**
> > >
> > > Dear Reviewer `1nxB`,
> > >
> > > We greatly appreciate your time and effort in reviewing our paper, as your feedback is incredibly valuable in helping us improve the quality of our paper.
> > >
> > > We gently invite you to revisit our previous responses as the discussion deadline is approaching, **with only 24 hours remaining**. We are eager to receive your response during this discussion period, and we would like to remind you that after **December 03 (AOE)**, we will not be able to answer any further questions you may have.
> > >
> > > In response to your suggestions, we have conducted comprehensive experiments. Specifically, we included：
> > >
> > > * Standard deviation in the retrieval tasks and the ablation study in Tables 1 and 8.
> > > * Theoretical computational complexity and computational time comparisons across baselines.
> > >
> > > We would greatly appreciate it if you could confirm whether our response satisfactorily addresses your concerns during this discussion phase.
> > >
> > > We understand you have a busy schedule, and we are genuinely grateful for your timely response. Your expertise and insights are critical to our work, and we deeply value your contributions.
> > >
> > > Thanks for your attention and best regards.

---

### Author Response · Authors · 2024-12-04
**Summary of Rebuttal Period**

We sincerely thank all PCs, SACs, ACs, and Reviewers for their time and effort in handling our submission #9785.

Overall, we received three reviews with positive ratings of 6 (`1nxB`), 6 (`N9hg`), and 6 (`1BNJ`).

**Paper strengths acknowledged by Reviewers:**

* **Novel approach and idea** (Reviewers `1nxB`, `1BNJ`).
* **Comprehensive experiments** effectively validate the effectiveness and versatility of Atomas. (Reviewers `1nxB`, `N9hg`, `1BNJ`)
* **Extensive ablation studies**, providing some insights into the molecule discovery domain (Reviewers `1nxB`, `N9hg`).
* **Intuitive and clear presentation and writing** (Reviewers `1nxB`, `N9hg`).

We have carefully addressed all concerns raised by the reviewers as follows:

* Adding additional standard deviation in the retrieval tasks and the ablation study in Tables 1 and 8.
* Including a theoretical computational complexity and providing computational time comparisons across baselines.
* Expanding discussion and including the more explicit rationale for the module, clarifying its differences from other works.
* Adding additional experiments on molecule-caption translation task to further demonstrate Atomas' generalization capabilities.
* Comparing more advanced baselines.


Once again, we deeply appreciate the reviewers' valuable insights and your efforts in managing the review process.

Best regards,

Authors of Submission #9785

---

### Meta-Review · Area_Chair_YYw4 · 2024-12-23

**Metareview:**

The paper proposes Atomas, a novel hierarchical molecular representation learning framework for molecule-text understanding and generation, leveraging fine-grained alignment at atom, fragment, and molecule levels. Its key strengths include a novel methodology for cross-modal molecule-text alignment, comprehensive experiments across 12 tasks demonstrating superior performance, and extensive ablation studies validating the model's robustness and scalability. Reviewers praised its clear presentation, insightful approach, and competitive results. However, weaknesses include concerns about limited novelty in some components compared to prior works, and a lack of additional datasets and baselines. The authors have done a good job to provide additional results for added baselines and benchmarks. The reviewers give scores 6,6,6 and unanimously lean acceptance of the paper.

**Additional Comments On Reviewer Discussion:**

Reviewers appreciated the paper's novel hierarchical alignment approach, strong experimental results, and clear presentation but raised concerns about missing standard deviations in results, computational efficiency, limited novelty in some components, and insufficient comparisons with advanced baselines and datasets. The authors addressed these by adding standard deviation results, providing computational complexity analysis, clarifying the novelty of their hierarchical alignment, and conducting additional experiments on the PubChem dataset to demonstrate generalization. They also updated comparisons with advanced baselines like ICMA and BioT5, effectively addressing most concerns and improving reviewer evaluations.

---

### Decision · Program_Chairs · 2025-01-22

Accept (Poster)